# Deep learning-based high-accuracy quantitation for lumbar intervertebral disc degeneration from MRI

Hua-Dong Zheng [1,2,9], Yue-Li Sun [3,4,5,9], De-Wei Kong[3], Meng-Chen Yin[3,5], Jiang Chen[6], Yong-Peng Lin[7], Xue-Feng Ma[8], Hong-Shen Wang[7], Guang-Jie Yuan[1,2], Min Yao[3,4,5], Xue-Jun Cui[3,4,5], Ying-Zhong Tian [1,2✉] & Yong-Jun Wang [3,4,5✉]

To help doctors and patients evaluate lumbar intervertebral disc degeneration (IVDD) accurately and efficiently, we propose a segmentation network and a quantitation method for IVDD from T2MRI. A semantic segmentation network (BianqueNet) composed of three innovative modules achieves high-precision segmentation of IVDD-related regions. A quantitative method is used to calculate the signal intensity and geometric features of IVDD. Manual measurements have excellent agreement with automatic calculations, but the latter have better repeatability and efficiency. We investigate the relationship between IVDD parameters and demographic information (age, gender, position and IVDD grade) in a large population. Considering these parameters present strong correlation with IVDD grade, we establish a quantitative criterion for IVDD. This fully automated quantitation system for IVDD may provide more precise information for clinical practice, clinical trials, and mechanism investigation. It also would increase the number of patients that can be monitored.

---

[1] School of Automation and Mechanical Engineering, Shanghai University, Shanghai 200072, China. [2] Shanghai Key Laboratory of Intelligent Manufacturing and Robotics, Shanghai 200072, China. [3] Longhua Hospital, Shanghai University of TCM, Shanghai 200032, China. [4] Spine Research Institute, Shanghai Academy of TCM, Shanghai 200032, China. [5] Key Laboratory of the Ministry of Education of Chronic Musculoskeletal Disease, Shanghai 200032, China. [6] Dongzhimen Hospital, Beijing University of Chinese Medicine, Beijing 100700, China. [7] Guangdong Provincial Hospital of Chinese Medicine, Guangzhou 510120, China. [8] Shenzhen Pingle Orthopedics Hospital, Shenzhen 518118, China. [9] These authors contributed equally: Hua-Dong Zheng, Yue-Li Sun. ✉email: troytian@shu.edu.cn; yjwang8888@126.com

Globally, as a major public health problem, low back pain has been the leading cause of disability worldwide for the past 30 years, with a burden on individuals, healthcare, and society[1]. IVD herniation, spinal stenosis, or ossification of the facets may pinch nerves, which may contribute to worsening LBP disease states. As an early pathological phenotype of LBP, IVD degeneration is necessary but hardly being quantified. IVD comprises a gel-like nucleus pulposus (NP), collagenous annulus-fibrosis (AF) layers, and ring-like cartilaginous endplates (EP), playing an important role in mechanical transmitting loads from body weight and daily activity through the spine column[2].

The extracellular matrix such as collagen and aggrecan provide tensile strength and osmotic-pressure regulation[3,4], which may degrade with aging. Accumulated compressive loads may accelerate this progressive process in IVD degeneration[5–7], which may lead to LBP with increased inflammation[8], nerve compression[9], and release of pain factors[4]. For IVD degeneration, T2-weighted (T2W) MRI is excellent at detecting the morphologic changes, including height loss and water-intensity loss. Pfirrmann et al. developed a grading system for IVD degeneration according to signal intensity and geometric features, which is one of the most accepted around the world[10]. However, it is highly dependent on the level of reader expertise, which showed only moderate interobserver agreement between readers in most studies[5], and the qualitative grading method cannot accurately reflect progressive changes in IVD-degeneration process[11]. Some measured IVD quantitative parameters were reported in early researches[12–17], which showed a better capacity to reflect the aging effect of IVD degeneration with relatively lower measurement error, thereby improving the quality of research on intervertebral disc degeneration[18]. However, the consistency and efficiency are not good enough to widely use, due to the need in manually segmenting the relevant area of IVDs and marking the corresponding feature points, as well as the inevitable subjective error and the limitation of the grayscale discrimination ability of human eyes. To some extent, researches on etiology and pathogenesis of IVD degeneration are progressing slowly with the limitation of measurement methods[16]. In addition, many studies reported the relationship between IVD height and demographic factors such as age and gender[19–23]. Although Shao et al.[17] established a linear model of IVD height-related parameters, the relationships between IVD height and other factors were not unified yet.

With the development of machine learning and deep learning, many studies regarded IVD-degeneration grading as a classification task. From the "shallow learning" task of manually making the degenerative features of the IVD[24–27], to the "deep learning" task of using the entire lumbar IVD bounding box to let the convolutional network learn the degenerative features by itself[28], the accuracy of these classification methods is comparable to that of radiology experts. However, these methods still require some manual input or complex detection algorithms, and hardly reflect progressive IVD-degeneration process. Meanwhile, many studies used manual segmentation methods to achieve quantitative analysis of the signal intensity and height characteristics of IVDs[13,14,16]. There has also been some studies on the quantitative measurement of intervertebral discs based on deep learning, but they did not use quantitative data to evaluate IVD degeneration, which may cause subjective bias and limit its clinical application[18,29]. Although the U-Net semantic segmentation model for the first time to achieve automatic segmentation and feature extraction of IVD-related regions[18], they did not reveal the IVD-degeneration process from their extracted geometric parameters.

Here, we developed an improved deeplabv3+ segmentation network with newly designed modules and a quantitative method for IVD degeneration. After evaluating model performance in segmentation accuracy, quantitation consistency, and application compatibility, a baseline characteristic of IVD signal intensity and geometric morphology among different gender and age and lumbar segments was established by extracting over 1000 MR images of a large patient population at different institutions in China, to develop a quantitative IVD-degeneration structured report. A diagram of this workflow is illustrated in Fig. 1.

## Result

**Segmentation-performance improvement with modelus (DFE, ST-SC, and MFF).** Figure 2 and Table 1 depict segmentation performance of the BianqueNet model with or without modelus (DFE, ST-SC, and MFF). The segmentation performance of deeplabv3+ network without any modelus showed a moderate accuracy, whose mDice and mIoU were 94.45% and 89.88% for the whole lumbar spine, 96.71% and 93.66% for vertebral body, and 94.38% and 89.43% for IVD. The segmentation performance of BianqueNet showed a better accuracy, whose mDice and mIoU were 94.70% and 90.35% for the whole lumbar spine, 97.03% and 94.25% for vertebral body, and 94.80% and 90.19% for IVD, indicating that segmentation performance of deeplabv3+ combined with the three modules (DFE, ST-SC, and MFF) has been improved significantly. Even in the smaller sample Data Set B, BianqueNet also showed good segmentation performance.

Notably, our model's segmentation contained more accurate and detailed structural information in IVDs and vertebral bodies in case 1 and case 2, while better discrimination in the boundary of IVD and vertebral body can be seen in case 3 (Fig. 2). The enhancement of the segmentation performance can significantly improve the accuracy of the corner detection for subsequent IVD-degeneration calculation, as shown in the feature-point part of Fig. 2.

**Segmentation performance in clinical sites with different magnetic field strength.** To test whether the model trained with MR images from Longhua Hospital (Data Set A) is applicable for other MR images from different hospitals, 60 MR images from other hospitals (Data Set C) and 80 MR images from Longhua Hospital (Data Set A) were randomly selected and segmented with the researcher and BianqueNet. Supplementary Table 1 depicts segmentation performance around these four hospitals. The segmentation performance for MR image from Dongzhimen Hospital was acceptably moderate, while those for other two hospitals showed no significant difference with the training set (Longhua Hospital).

**Quantitation performance in different MR images with different resolutions.** A total of 230 IVDs and 276 vertebral bodies of 46 subjects were segmented after resolution had been adjusted from 320*320 to 512*512. The results showed a good consistency in using different parameter-calculation algorithms for MR images with different resolutions. Among them, the normalized IVD geometric parameters (DHI and HDR) have extremely high ICC values, which are 0.958 ($p = 0.000$) and 0.956 ($p = 0.000$), respectively, while the normalized IVD signal intensity ($\Delta SI$) showed high ICC value of 0.874 ($p = 0.000$), as shown in Supplementary Table 2. The reason why we set MR image resolution of 512*512 for final model input and model A for the final model is because a large proportion of images present resolution of 512*512 among all collected MR images. Considering that using interpolation method may miss or change information from the image by reducing or enlarging image sizes, we finally chose the middle-image resolution of 512*512 to retain the original information of MR images at the most extent.

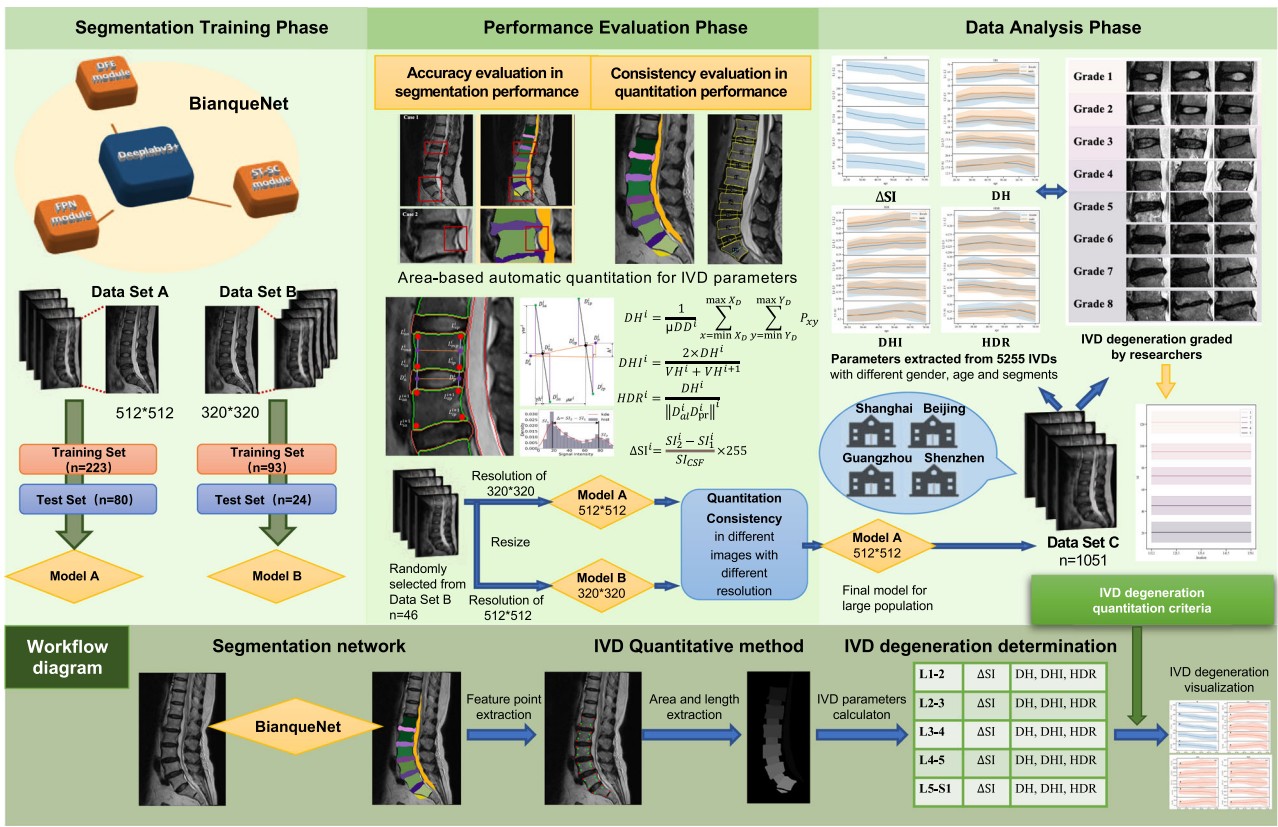

**Fig. 1 A flowchart of the study process from training and testing phase to data analysis phase with BianqueNet.** Mid-sagittal T2W lumbar MR images exported into two different resolutions (512*512, 320*320) are used to train and test two models (Model A, Model B). After segmentation accuracy evaluation, IVD parameters are quantified based on IVD-related area. As Model A shows good performance on quantitation consistency for MR images with various resolutions, it is used to establish baseline characteristic with different IVD degeneration among 5255 IVDs. Workflow diagram at the bottom presents the segmentation network, IVD quantitation method, and IVD degeneration determination.

**Comparison with automatic quantitation and manual measurement.** For comparison, the mean values of DH, DHI, and HDR were set as control to test model quantitation performance (Table 2). Thanks to carefully measuring all the IVDs by two senior residents, the intraobserver agreement between the two residents' measurements on the 75 IVDs in 15 MR images presents ICC value of 0. 944 for DH (95% CI: 0.912, 0.964), 0.913 for DHI (95% CI: 0.862, 0.945), and 0.881 for HDR (95% CI: 0.730, 0.939), indicating a good interobserver agreement in all the IVD-related area measurements and index calculation between the two senior residents. Subsequently, mean of their measurements, as a control, was used to compare with the results extracted by the proposed network. There was moderate-to-good intraobserver agreement between machine and manual measurements with ICC value of 0.954 for DH (95% CI: 0.928, 0.971), 0.908 for DHI (95% CI: 0.856, 0.941), and 0.917 for HDR (95% CI: 0.810, 0.957).

Compared with clinical radiologists and residents, the model provided highly repeatable and accurate IVD geometric measurements. In particular, the consistency in VB area-measured results between model and residents was the highest (ICC: 0.964), which may accord with segmentation validation in these areas (Fig. 2). The consistency in IVD area-measured results between model and residents was relatively lower because area for average IVD height calculation was selected with residents' own discrimination (Supplementary Fig. 1), while it was detected with featured points by model calculation (Fig. 6h). For calculated parameters (DH, DHI, and HDR), both measurement error and selection subjectivity may affect consistency between model and residents, but our evaluation result still presents acceptable good performance (ICC > 0.9).

**Model performance in patient subgroups by gender, age, and segments.** After screening 1508 MRI images in 4 sites around China, a total of 1051 individuals were collected, in which there are 73 excluded for unaligned outlines (diagnosed as lumbar spondylolisthesis), 45 excluded for abnormal signal intensity distribution (diagnosed as spine tumors), 364 excluded for irregular structures (diagnosed as IVD herniation or vertebral body ossification), and 144 excluded for imaging quality (segmentation results and corner detection did not meet the requirements of parameter calculation). The demographic information (age and gender) distributed evenly as shown in Supplementary Table 3, which was integrated to conduct correlation analysis with IVD parameters.

Supplementary Fig. 2 and Table 3 show comprehensive baseline characteristics of IVD parameters in a larger population. $\Delta SI$ in IVDs decreased with age, while DH of IVDs increased with age, reaching peak at the age of 50–60 ($P < 0.01$). There is no significant difference between male and female in $\Delta SI$ in IVDs, while DH, DHI, and HDR of IVDs were significantly higher in males than those in females ($P < 0.01$). In addition, DH, DHI, and HDR were significantly higher in lower segmental IVDs (L3–L4, L4–L5 and L5–S1) than upper ones (L1–L2 and L2–L3), and disc height of L4–L5 IVDs was the highest ($P < 0.01$). Through multivariate linear-regression analysis (MLR), we investigated the distribution of IVD geometric parameters and signal intensity of each segment in a large population with different age and gender (shown in Table 5). Variables such as gender and segments have significant correlations with $\Delta SI$, while variables such as age, gender, and segments have significant correlations with geometric parameters.

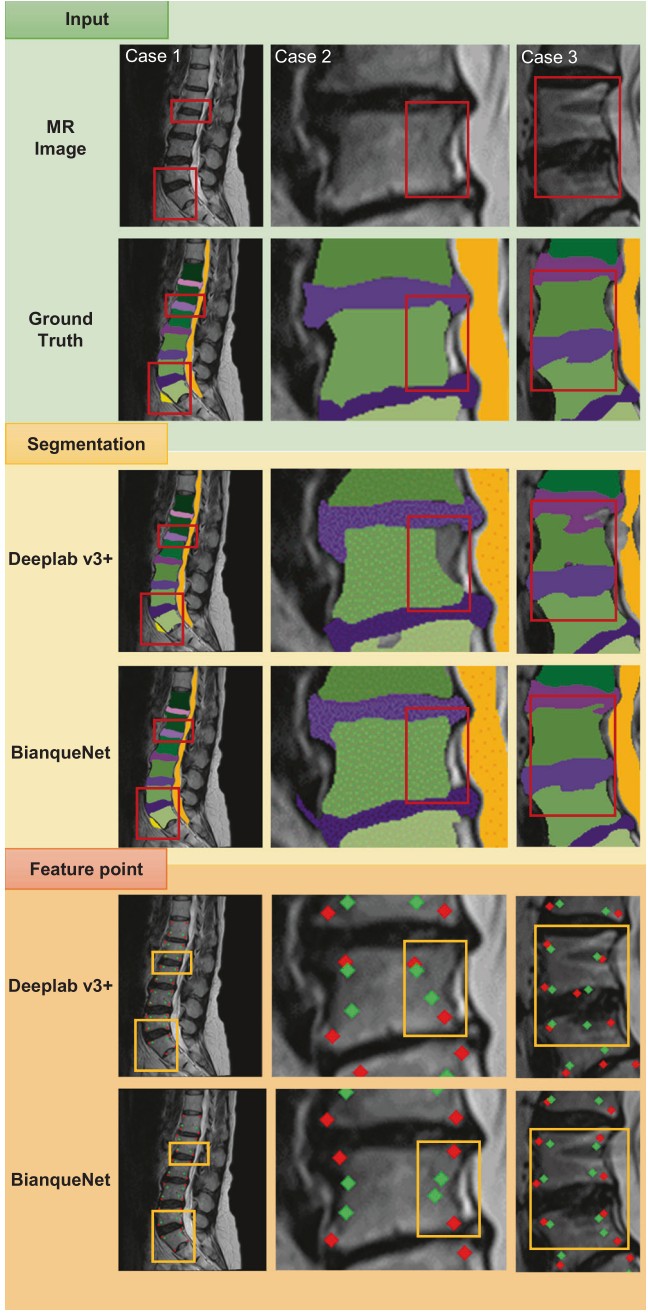

**Fig. 2 The segmentation performance of BianqueNet in three typical cases and the influence of different segmentation accuracy on feature-point detection and calculation.** Segmentation results in case 1 and case 2 indicate that detailed information on the boundaries of vertebral bodies and IVDs is hard to detect. Segmentation in case 3 shows that irregular boundaries between the IVD and the vertebral bodies may interrupt segmentation with slight structural lesions or imaging defects. Feature-point-extracted results indicate that the precise segmentation may significantly improve the corner detection on vertebral bodies (red dots), thereby affecting the calculation results of the characteristic points on IVDs (green dots).

**Validity in IVD-degeneration grading performance**. Considering height decrease and water-content loss with IVD degeneration, a regression analysis was conducted to investigate the correlation between IVD parameters and degeneration grading ($\Delta SI$ with corresponding grading (1, 2, 3, 4, and (5–8)), geometric

parameters (DH, DHI, and HDR) with corresponding grading ((1–5), 6, 7 and 8)) in each segment. As shown in Table 4, IVD parameters showed a good accordance to the modified Pfirrmann grade.

The result from a stronger correlation between the modified Pfirrmann grade (1, 2, 3, 4, and (5–8)) and $\Delta SI$ ($\rho = -0.966$, $P = 0.000$), which demonstrates that the water content of NP is decreasing with the whole IVD-degeneration process. Therefore, specific ranges of $\Delta SI$ according to the modified Pfirrmann grade (1, 2, 3, 4, and (5–8)) were calculated and set as automatic grading criteria, which are shown in Fig. 3a and Supplementary Table 4.

According to these statistical results, ranges of IVD geometric parameters and levels of IVD signal intensity for each segment in these participates of different ages and genders were established as Chinese population baseline, which is shown in Fig. 3 and Supplementary Tables 5–8.

Although our method is focused on quantitative measurement other than degeneration-grade classification, it still presents strong accordance with manual IVD-degeneration grading (macroF1: 92.02% and 90.63% in two data sets) by means of quantifying IVD degeneration, which is shown in Table 5.

## Discussion

In this work, we propose an automatic IVD-degeneration quantitative method based on deep-learning segmentation, in which a powerful semantic segmentation network (BianqueNet) was designed to achieve accurate segmentation with IVD-related areas from T2W MR images. In the quantitation section, an improved histogram method was proposed, and automatic calculation methods were modified to qualify signal intensity and geometric information of vertebral bodies and IVD. To investigate baseline characteristics of IVD, this method was used in a large population to collect IVD geometric parameters (structural collapse) and signal intensity (water content) with different degeneration grade, age, and gender. A IVD-degeneration quantitative criteria in different population subgroups were finally established by correlation analysis and multiple-regression analysis. Finally, the deviation method was used to achieve the degeneration grading and quantitative analysis on IVDs.

The deep-learning approach allows the network to perform lumbar IVD segmentation and parameter quantitation simultaneously, which may help doctors and patients obtain more IVD-degeneration status information from traditional T2W MR images. Considering large time-consuming and high internal/external differences of IVD manual measurement shown in previous studies, this approach may provide a relatively efficient and accurate solution in a large population to extract more consistent IVD parameters and benefit several clinical applications (such as preventive screening, therapeutic evaluation, decision secondary, and mechanism investigation).

To provide a valuable diagnosis tool for IVD degeneration, quantitative-analysis methods may improve currently used qualitative classification methods[15]. Although many IVD quantitative methods were proposed and developed, the measured IVD parameters were not accepted and used with limited reliability and validity[12,13,15,30]. In this work, based on previous studies, some appropriate improvement was designed in signal intensity and geometric information, achieving automatic IVD parameter extraction by means of the latest deep-learning and image processing techniques. The precise segmentation of IVD-related areas is a key step to achieve the automatic extraction of IVD parameters. The proposed DFE and MFF modules integrate multi-scale-rich semantic information to improve the capacity of scene analysis, which may improve the ability to distinguish boundaries between vertebral bodies and IVDs. Meanwhile, the

**Table 1 Segmentation performance of BianqueNet compared with those without other modules (DFE, ST-SC, and MFF) in test sets.**

| Model | Module | | | Vertebral body | | IVD | | Lumbar spine | |
|---|---|---|---|---|---|---|---|---|---|
| | DFE | ST-SC | MFN | mDice | mIoU | mDice | mIoU | mDice | mIoU |
| DeepLabv3+ | | | | 0.9671 | 0.9366 | 0.9438 | 0.8943 | 0.9445 | 0.8988 |
| DeepLabv3+ + DFE | √ | | | 0.9681 | 0.9384 | 0.9444 | 0.8960 | 0.9455 | 0.9006 |
| DeepLabv3+ + DFE + ST-SC | √ | √ | | 0.9692 | 0.9405 | 0.9458 | 0.8982 | 0.9468 | 0.9028 |
| DeepLabv3+ + DFE + ST-SC + MFF (BianqueNet) | √ | √ | √ | 0.9703 | 0.9425 | 0.9480 | 0.9019 | 0.9470 | 0.9035 |
| BianqueNet[a] | √ | √ | √ | 0.9599 | 0.9255 | 0.9310 | 0.8717 | 0.9345 | 0.8832 |

[a]Means MR images were from Data Set B, others were from Data Set A, all MR images of the training model were from Longhua Hospital, Shanghai University of TCM.

**Table 2 Consistency analysis of IVD parameters measurements between model and residents.**

| Intraclass correlation[a] | | Model vs residents | | Resident A vs resident B | |
|---|---|---|---|---|---|
| | | ICC[b] | 95%CI | ICC[b] | 95%CI |
| IVD geometric measurements | VB area | 0.964[***] | (0.933, 0.979) | 0.952[***] | (0.912, 0.972) |
| | IVD area | 0.934[***] | (0.785, 0.971) | 0.916[***] | (0.763, 0.961) |
| | DH | 0.954[***] | (0.928, 0.971) | 0.944[***] | (0.912, 0.964) |
| | DHI | 0.908[***] | (0.856, 0.941) | 0.913[***] | (0.862, 0.945) |
| | HDR | 0.917[***] | (0.810, 0.957) | 0.881[***] | (0.730, 0.939) |

[a]Type A intraclass correlation coefficients using an absolute agreement definition.
[b]The estimator is the same, whether the interaction effect is present or not. Two-way mixed effects model where people effects are random and measures effects are fixed. ICC, intraclass correlation coefficient; 95% CI, 95% confidence interval. ***, $P < 0.001$.

**Table 3 The results of multiple regression analysis of △SI, DH, DHI, HDR and gender, different ages, and different segments.**

| measure-ment | △SI | | DH | | DHI | | HDR | |
|---|---|---|---|---|---|---|---|---|
| | Coef. | $P > \lvert t \rvert$ | Coef. | $P > \lvert t \rvert$ | Coef. | $P > \lvert t \rvert$ | Coef. | $P > \lvert t \rvert$ |
| female | −0.0279 | 0.141 | −0.2541 | 0.000 | −0.1121 | 0.000 | 0.1115 | 0.000 |
| male | 0.000 | / | 0.000 | / | 0.000 | / | 0.000 | / |
| 20–30 | 0.000 | / | 0.000 | / | 0.000 | / | 0.000 | / |
| 30–40 | −0.1669 | 0.000 | 0.0796 | 0.003 | 0.0557 | 0.057 | 0.1100 | 0.000 |
| 40–50 | −0.3802 | 0.000 | 0.1110 | 0.000 | 0.0927 | 0.001 | 0.0980 | 0.001 |
| 50–60 | −0.4826 | 0.000 | 0.1612 | 0.000 | 0.1577 | 0.000 | 0.0440 | 0.118 |
| 60–70 | −0.6002 | 0.000 | 0.1427 | 0.000 | 0.1687 | 0.000 | 0.0099 | 0.730 |
| 70–90 | −0.5137 | 0.000 | 0.0328 | 0.120 | 0.0806 | 0.000 | −0.0674 | 0.006 |
| L1-L2 | 0.2800 | 0.000 | −0.7181 | 0.000 | −0.6708 | 0.000 | −0.4932 | 0.000 |
| L2-L3 | 0.1719 | 0.000 | −0.3832 | 0.000 | −0.4155 | 0.000 | −0.2912 | 0.000 |
| L3-L4 | 0.0907 | 0.000 | −0.1593 | 0.000 | −0.1942 | 0.000 | −0.1122 | 0.000 |
| L4-L5 | 0.000 | / | 0.000 | / | 0.000 | / | 0.000 | / |
| L5-S1 | 0.1526 | 0.000 | −0.0520 | 0.023 | −0.0312 | 0.206 | 0.1105 | 0.000 |
| Adj R² | 0.4191 | | 0.4739 | | 0.3872 | | 0.3078 | |

95% confidence interval; Multivariate regression analysis of standardized regression coefficients.

**Table 4 Correlations between IVD parameters and modified Pfirrmann grading.**

| lumbar level | △SI | DH[a] | | DHI | | HDR | |
|---|---|---|---|---|---|---|---|
| | | female | male | female | male | female | male |
| L1/L2 | −0.966* | −0.421* | −0.296* | −0.304* | −0.235* | −0.473* | −0.397* |
| L2/L3 | | −0.481* | −0.417* | −0.354* | −0.398* | −0.575* | −0.455* |
| L3/L4 | | −0.639* | −0.470* | −0.530* | −0.443* | −0.626* | −0.539* |
| L4/L5 | | −0.656* | −0.696* | −0.560* | −0.665* | −0.709* | −0.758* |
| L5/S1 | | −0.701* | −0.687* | −0.641* | −0.664* | −0.744* | −0.778* |

$\rho$, Spearman rank correlation coefficients.
*at the $p < 0.01$ level (two-tailed), the correlation is significant.
[a]DH is the only parameter that is not standardized, while △SI can be applied to MRI at different centers, and DHI and HDR can be applied to different types of imaging means and physical measurements.

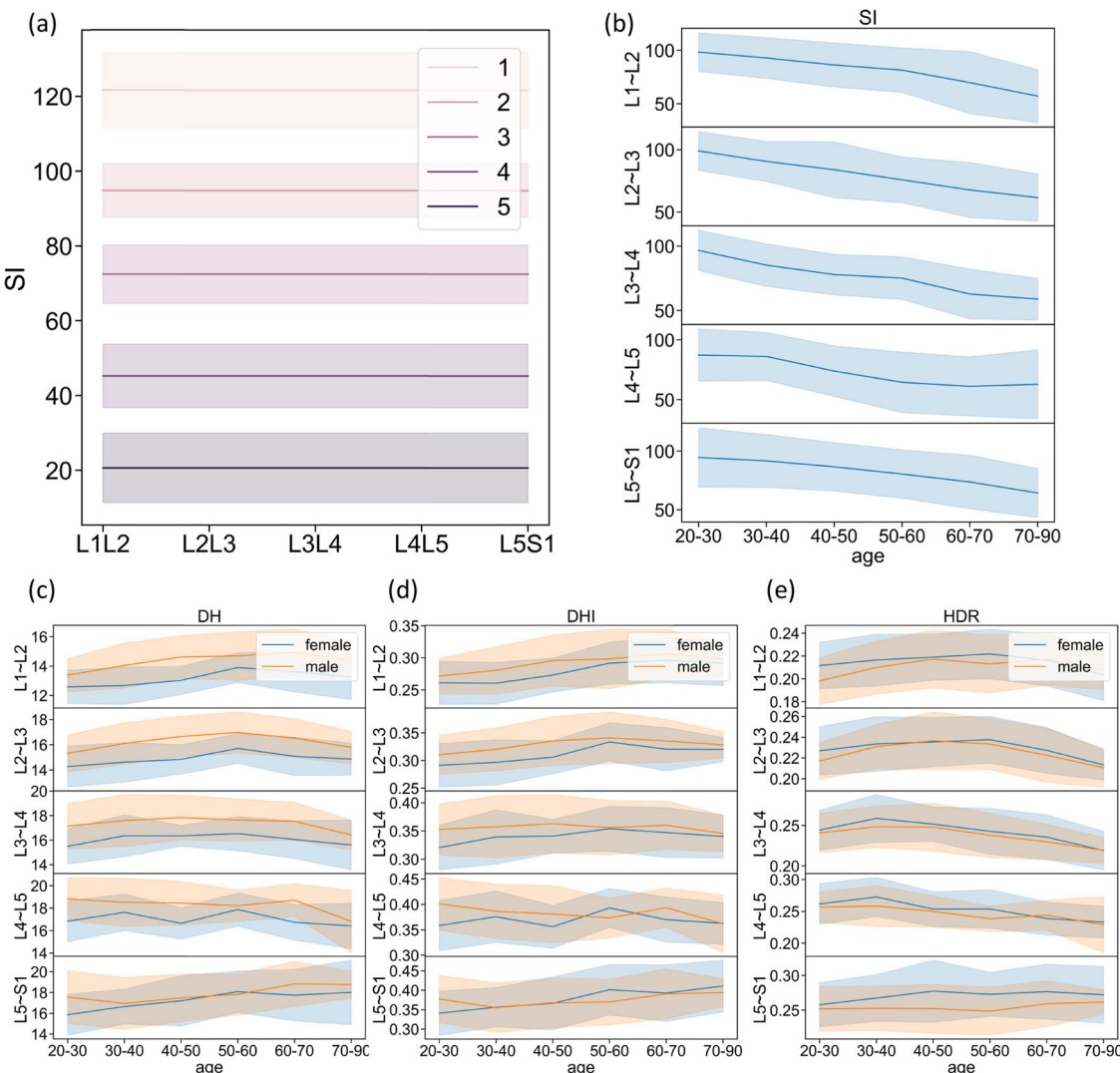

**Fig. 3 Baseline characteristics of IVD parameters in geometric and signal intensity.** The mean and standard deviation ($\sigma$) of the $\Delta$SI of each of the modified Pfirrmann grading system (levels 1, 2, 3, 4, and 5) were calculated from Dataset C, which is used to quantify IVD degeneration (**a**), $\Delta$SI (**b**), DH (**c**), DHI (**d**), and HDR (**e**) were quantified in different age, gender, and segments to establish population baseline.

**Table 5 Accuracy of IVD degeneration grading with $\Delta$SI in IVD.**

| Modified Pfirrmann grade | | 1 | 2 | 3 | 4 | 5–8 | macro-average (%) | macroF1(%) |
|---|---|---|---|---|---|---|---|---|
| Data set A | Precision (%) | 60.76 | 97.28 | 99.40 | 97.89 | 89.08 | 88.89 | 92.02 |
| | Recall (%) | 100 | 90.96 | 97.84 | 90.05 | 98.15 | 95.40 | |
| Data set B | Precision (%) | / | 81.82 | 93.55 | 100 | 85.71 | 90.27 | 90.63 |
| | Recall (%) | / | 90.00 | 90.63 | 83.33 | 100 | 90.99 | |

proposed ST-SC module can increase the focus on target information in different spatial domains, providing more accurate and fine-grained structural information for the upsampling path, to enhance the model obtaining more accurate and detailed contour information of IVDs and vertebral bodies. Notably, the trained model has been validated among additional data sets from other three hospitals, demonstrating that BianqueNet may perform consistency segmentation on MR images from different machines.

A normalized processing was added to improve general performance and application, by optimizing the IVD histogram

analysis method. Benefiting from powerful segmentation performance of BianqueNet, our proposed IVD quantitative method may be able to precisely detect angle points of vertebral bodies and rapidly calculate feature points of IVDs, thus achieving an automatic IVD area-based quantitation in the first time. In addition, consistency evaluation indicated no significant difference between automated approach and senior radiologists/orthopedic residents' measurements.

The normalized $\Delta SI$ showed excellent linear correlation with IVD degeneration (R = −0.966, P = 0.000), suggesting that IVD histogram analysis is a suitable tool for objective and continuous

IVD-degeneration classification, which is similar to the conclusion of Waldenberg et al.[13] Thus, we statistically analyzed IVD characteristic in ΔSI with different degeneration grades (corresponding to the modified Pfirrmann grade system), in which the results showed that the histogram features of IVD signal intensity had strong applicability in IVD-degeneration grading.

As IVD-structure collapse status is another important reference in IVD-degeneration grading, we carried out a correlation analysis between IVD geometric parameters (DH, DHI, and HDR) and IVD-structure collapse status. HDR presents the strongest linear correlation with IVD structural collapse, due to its containing both height and shape information that may be the main references in manual degeneration grading. In contrast, DHI showed limited correlation with manual grading, for the height of vertebral bodies used in DHI calculation is not likely considered in manual degeneration grading according to both Pfirrmann grade system and its modified version. Considering the variability of individual IVD height, it may affect the overall linear correlation between the geometric parameters and IVD-degeneration grades in structural collapse.

We further investigated the relationship between IVD quantitative parameters (DH, DHI, HDR, and ΔSI) and baseline demographic information (age, gender, and segment) with a multiple-regression analysis. IVD signal intensity (ΔSI) showed a stronger correlation with age and segment, indicating that accumulated loads may lead to water-content loss in IVD. On the other hand, height of IVD presents a phenomenon of increasing in young age and decreasing in older age, which accorded with previous studies[23]. However, our study revealed that there are no linear relationships between IVD geometric parameters (DH, DHI, and HDR) and age, which showed disagreement with previous studies[19]. For the DH parameter, the influence of gender is greater than that of the age. For the normalized parameters DHI and HDR, the influence of gender and age seems similar. The position of the structure shows the greatest influence on the geometric parameters. In fact, Pfirrmann grade system was designed based on symptomatic patients with an average age of about 40 years old[19], whose reliability for early IVD degeneration or IVD degeneration in the elderly people may be unsatisfied. In our study, these correlation results played important roles in IVD quantitative degeneration-criterion establishment, achieving automatic quantitation for IVD degeneration in asymptomatic patients of different ages.

An important distinction of our study from previous works is the accurate IVD-parameter extraction from MR images of a large population to establish a criterion of IVD-degeneration quantitation, by means of a powerful segmentation network (BianqueNet) and improved area-based calculation.

Regarding future clinical practice and assessment, we will insert this network into MR-image system (Computer Software Intellectual Property Right, National Copyright Administration, P.R. China, No. 2021SR1211447) and export a structural lumbar intervertebral disc degeneration report like Supplementary Fig. 3 and Video 1 for doctors, patients, and researchers. Compared with traditional text-description MRI report, our quantitative report may provide more accurate IVD parameters to reflect height collapse and water-content loss with IVD degeneration. According to IVD baseline characteristic criteria in each age, gender, and segments, deviation of IVD geometric parameters and Pfirrmann grade based on signal intensity will be obtained automatically to reflect both structural collapse status and water-content loss in IVD comprehensively, which may provide more precise information for clinical practice (lumbar MR-image structural report), clinical trials (efficacy assessment), and mechanism investigation (biomechanics research and finite-element analysis). Notably, these baseline characteristics will be updated dynamically as these MR-image data are collected and summarized.

In our result, we found that the turning point for "peak IVD height" is in the 50–60 age range, which may be a secondary degenerative process. Due to the changes in vertebral osteoporosis, the endplate of the vertebral body becomes more depressed, which may make IVD sink into the vertebral body, resulting in lower vertebral height and higher disc height[20,31].

Also, there are many studies on the relationship between age and height of IVD and vertebral bodies. H.S. Monoo-Kuofi et al.[32] concluded that IVD height increases with age, but not in a linear fashion, with alternating periods of overgrowth and thinning, and a significant decrease of 2.5% after age 50. These studies support our results. In the future, we will continue to collect data from more MRI and possibly investigate these IVD parameters as aging.

Our study has some limitations. First, this retrospective study may be subject to potential selection bias. Some prospective studies should be rigorously conducted to test the clinical utility of this proposed model. Second, our deep learning model was trained and tested using Chinese patients, so its reproducibility among different ethnic people should be further evaluated. In future, it will be important to combine radiomics and prospective design and integrate all kinds of clinical examination, fluid-flow biomechanics, and molecular approaches to improve accuracy in IVD-degeneration evaluation.

In conclusion, we present a fully automated deep-learning-based lumbar-spine segmentation network and an area-based quantitative method to evaluate IVD degeneration according to the extracted parameters from a large population. Our approach can be used to improve IVD-degeneration evaluation with high accuracy and consistency.

## Methods

**Patients and datasets.** This study was approved by the Institutional Review Board (IRB) in all the participating sites. Written informed consent was waived because of the retrospective nature of the data collection (age/gender) and the use of deidentified MR images.

Two separate segmentation models were trained and tested among mid-sagittal T2 lumbar-spine images of different resolution (512*512 in Data Set A and 320*320 in Data Set B). All the subjects' lumbar-spine MR imaging was included in the Longhua Hospital, Shanghai University of TCM between January 1, 2019, and December 31, 2020, among which there are 223 subjects using a 1.5-T MRI unit (MAGNETOM Aera XJ, SIEMENS, Data Set A) and 63 subjects using another 1.5-T MRI unit (MAGNETOM Avanto, SIEMENS, Data Set B). These MR images were exported and randomly allocated into each training set or test set (Fig. 1). All images were labeled by LabelMe (version 3.3.6, CSAIL, Massachusetts Institute of Technology)[33]. Based on the structural features mentioned in the modified Pfirrmann grading system, the segmentation area of 14 parts included 5 vertebral bodies (L1–L5), 5 lumbar IVDs (L1/L2–L5/S1), sacrum (S1), presacral fat area, cerebrospinal fluid area (CSF) in the spinal canal, and background as Fig. 4b. Segmentation performance of IVD-related areas was tested by the mean Dice coefficients (mDice) and mean Intersection over Union (mIoU).

In order to establish lumbar IVD baseline data in a large population, Data Set C composed of 1051 mid-sagittal T2 lumbar-spine images with different age (20–90) and gender was used to extract data by segmentation model and quantitative method among four hospitals around China, including Longhua Hospital, Shanghai University of TCM, Guangdong Provincial Hospital of Chinese Medicine, Shenzhen Pingle Orthopedics Hospital, and Dongzhimen Hospital, Beijing University of Chinese Medicine between January 1, 2019 and March 30, 2021. The imaging parameters of all sites are summarized in Supplementary Table 9.

## Proposed network model

*Overview of the BianqueNet architecture.* As presented in Fig. 4, mid-sagittal T2W lumbar MR images are input into the backbone (a resnet101 network[34] that uses the atrous separable convolution to improve the last stage) for 16 times the down-sampling, from which richer semantic information and more dense features are extracted through the depth feature extraction module (see Depth feature extraction section for details). To restore more detailed features of segmentation targets, the original quadruple upsampling operation in Deeplabv3+[35] was modified with double upsampling, while the general bilinear interpolation was replaced to transpose convolution for upsampling. At the same time, input the feature maps of

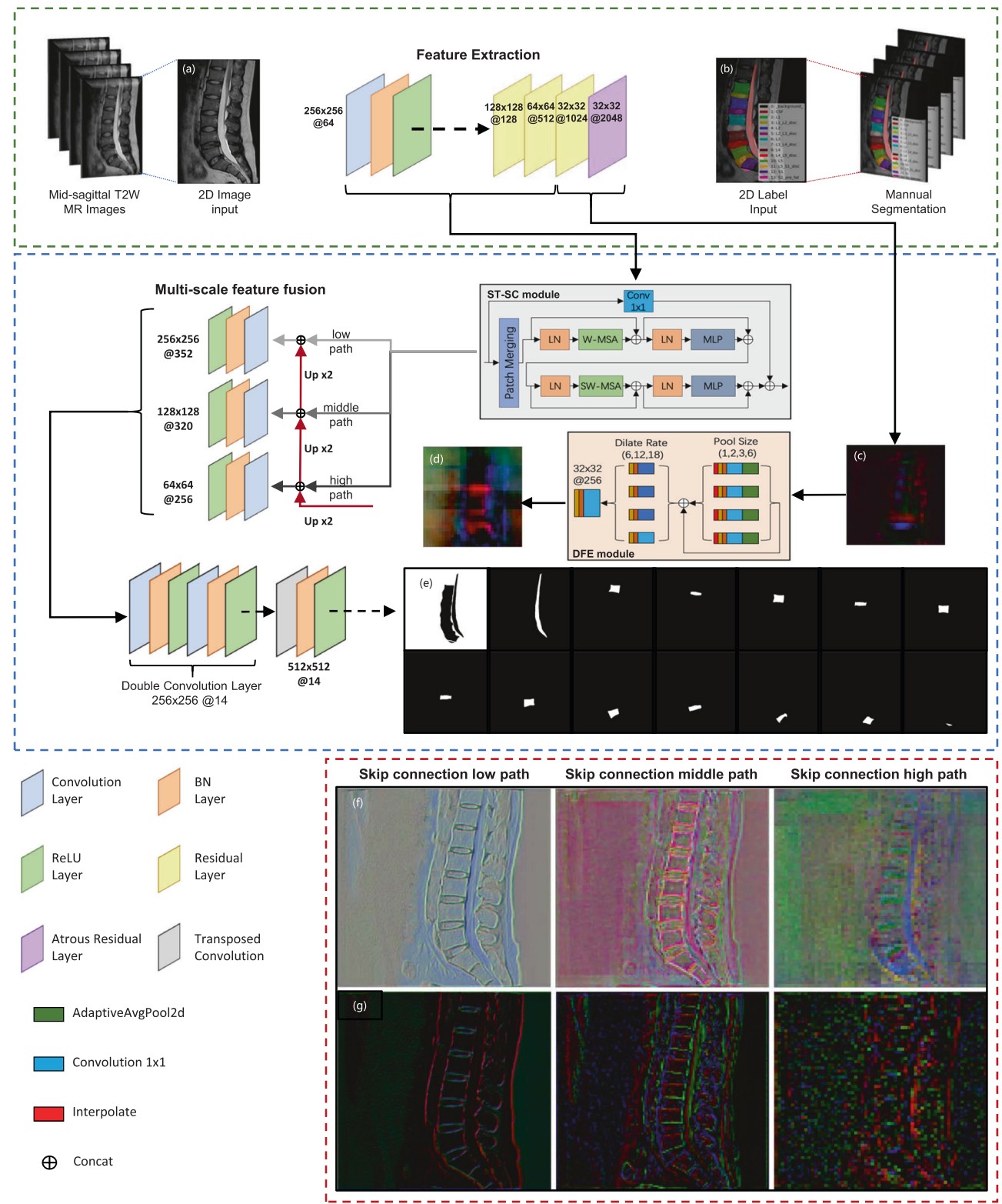

**Fig. 4 The proposed BianqueNet consisted of three innovative modules. a** Input MRI, (**b**) annotations, feature map before (**c**) and after (**d**) DFE module, and (**e**) each image-channel output by the model corresponds to a segmentation area. Feature map of different skip-connection path with (**f**) and without (**g**) ST-SC module.

different resolutions obtained by downsampling to the Swin Transformer-skip connection module (see "Swin Transformer–skip connection" section for details), and then fuse the upsampled feature maps of the same resolution to obtain feature maps of different scales. According to Feature Pyramid Network[36], multi-scale feature fusion module (MFF) is used to combine the feature maps with strong low-resolution semantic information and feature maps with weak high-resolution

semantic information but rich spatial information. Then a 3*3 double-convolutional layer is used for the fused feature map to improve the feature, and finally a double upsampling operation is performed to obtain a dense prediction image.

*Swin transformer–skip connection module.* Applications with transformer-based vision backbones such as Vision Transformer (ViT) achieved innovative

technological breakthrough in recent years[37,38], in which Swin Transformer, a layered transformer based on shifted windows, makes it compatible with a broad range of vision tasks. Compared with earlier sliding-window-based self-attention approaches[39,40], Swin Transformer performs higher efficiency and lower complexity. In this study, a skip-connection module was designed with two successive Swin Transformer blocks, called as ST-SC. As shown in Fig. 4, the Swin Transformer block is consisting of a shifted-window-based multi-head self-attention (MSA) module and a 2-layer multi-layer perceptron (MLP) with GELU nonlinearity. A layer norm (LN) layer is applied before each MSA and MLP module, while a residual connection is applied after each module[37]. At the same time, a 1*1 convolutional layer is applied after two successive Swin Transformer blocks, and finally the two output features are spliced to provide more accurate and fine-grained structural information for upsampling. To avoid affecting dense-feature output from upsampling, the number of output channels for the ST-SC module is adjusted to 1/8 times that of the downsampled feature maps. Calculation formulas are shown as the following:

$$\overline{SC}_0 = W\text{-}MSA(LN(X)) + X \tag{1}$$

$$SC_0 = MLP\left(LN\left(\overline{SC}_0\right)\right) + \overline{SC}_0 \tag{2}$$

$$\overline{SC}_1 = SW\text{-}MSA\left(LN\left(SC_0\right)\right) + SC_0 \tag{3}$$

$$SC_1 = MLP\left(LN\left(\overline{SC}_1\right)\right) + \overline{SC}_1 \tag{4}$$

$$Y = \varnothing_f(\hat{X}, SC_1) \tag{5}$$

where $X$ and $\hat{X}$ denote the down-sampled feature maps of different resolutions and their output of 1*1 convolution, respectively; $\overline{SC}$ and $SC$ denote the output features of the $(S)W$-MSA module and the MLP module, respectively; W-MSA and SW-MSA denote window-based multihead self-attention using regular and shifted-window partitioning configurations, respectively[37]. $\varnothing_f$ is the feature-fusion function, and $Y$ is the output of ST-SC module.

Compared with segmentation by deeplabv3+ without ST-SC module, the contour information of the vertebral body in the feature map of skip-connection middle path is more accurate, and the contour information of intervertebral disc and cerebrospinal fluid in the feature map of skip-connection high path is more accurate (Fig. 4f, g). Therefore, the ST-SC module may provide more accurate detailed information for the upsampling path by increasing the focus on target information in different spatial domains.

*Depth feature extraction module.* In this study, the depth-feature extraction module (DFE) is designed between backbone output section and upsampling sections. Feature-map output from the backbone is extracted-feature information of different depth through pooling operation with different scale in the pyramid pooling module[41]. Combining the fused global feature map with the backbone–output-feature map, a multiscale contextual information feature map of 4096 channels was obtained to further extract a dense semantic feature map of 256 channels through the atrous spatial pyramid pooling (ASPP) module[35].

*Weighted dice-loss function.* A weighted dice-loss function as below was proposed to enhance segmentation performance by estimating the difficulties in different images with typical or atypical structure, which ensured consistency in segmentation:

$$L_{wdice} = \frac{1}{C}\sum_{j=1}^{C}\xi_j\left(1 - \frac{2\sum_{i=1}^{N}p_{1i}g_{1i}}{2\sum_{i=1}^{N}p_{1i}g_{1i} + \sum_{i=1}^{N}p_{0i}g_{1i} + \sum_{i=1}^{N}p_{1i}g_{0i}}\right) \tag{6}$$

This formula was used in the output of the SoftMax layer, where the $p_{1i}$ is the probability of voxel $i$ (target) and $p_{1i}$ is the probability of voxel $i$ (nontarget). So was for $g_{1i}$ and $g_{0i}$. $j$ represents different segmentation areas, $C$ represents the total number of channels, which is taken as 14. $\xi$ represent the weight of different segmentation channels. According to the experimental analysis results, channel weight was set to 0.9, 0.8, and 1 for vertebral body, IVD, and the other, respectively, which may achieve the best segmentation performance.

For avoiding that the subsequent feature-extraction operations are affected, corrosion and expansion operations were used to remove the burrs (Fig. 4e).

## Lumbar IVD quantitative analysis
*Parameter calculation based on IVD-related area segmentation.* Based on previous studies[18,25–27], signal-intensity difference (ΔSI) in IVD areas was used to quantify the blurring degree of boundary between NP and AF, which indicates water-content loss in IVD degeneration. Average disc height (DH), disc-height index (DHI), and disc height-to-diameter ratio (HDR) were used to quantify structural collapse with IVD degeneration (Fig. 5). Detailed quantitative methods were described as below.

*Signal-intensity histogram features.* The histogram feature is used to quantify different signal-intensity distribution in different areas from MRI, in which the X axis represents different signal intensities, and the Y axis represents the corresponding

number of pixels. A two-peak distribution has been analyzed in healthy IVD from MRI, because the sharpness of the boundary between the NP and the AF can be well characterized with large amounts of pixel with two major signal intensities (Fig. 5d)[13]. With IVD degeneration, water-content loss in NP can be measured in histogram-feature distribution changes, which presents that previous higher signal intensity (light) in the IVD area gradually becomes lower (dark) (Fig. 5g). The difference in pixel numbers corresponding to different signal intensities can well describe the degeneration state. To reduce the influence of individual differences and MR-imaging condition, a modified method was used in calculating the difference between the two peaks in IVD signal intensity histogram after being normalized with the peak signal intensity of CSF in the spinal canal (Fig. 5f). The calculation formula of the signal-intensity difference (ΔSI) between two peaks is shown as the following:

$$\triangle SI^i = \frac{SI_2^i - SI_1^i}{SI_{CSF}} \times 255 \tag{7}$$

Among them, $SI_1^i$ and $SI_2^i$ respectively represent the signal-intensity values corresponding to the 1st and 2nd peaks of the histogram of the IVD, $i$ represents the position of the $i$th IVD. $SI_{CSF}$ represents the signal intensity corresponding to the peak of the histogram of the CSF area, and 255 is an amplification factor.

*Vertebral body height.* According to the channels of the segmented vertebral body, the Shi–Tomasi corner detection method was used to accurately point the four corner vertices (superior–anterior ($L_{sa}^i$), superior–posterior ($L_{sp}^i$), inferior-anterior ($L_{ia}^i$), and inferior-posterior ($L_{ip}^i$)) of the vertebral body (Fig. 5h). The Euclidean distance between two midpoints ($L_{ma}^i$ of $L_{sa}^i$ and $L_{ia}^i$, $L_{mp}^i$ of $L_{sp}^i$ and $L_{ip}^i$) is defined as the vertebral body diameter (Fig. 5h). The vertebral body diameter (VD) calculation formula is shown as the following:

$$VD^i = \sqrt{\sum_{j=1}^{2}\left(L_{maj}^i - L_{mpj}^i\right)^2} \tag{8}$$

where $i$ denotes the $i$th vertebral body, in the range from 1 to 5, $j$ denotes the midpoint coordinate dimension, values of 1,2.

The area of the vertebral body was calculated with the sum of all the pixel values of the vertebral body mask channel, and then the vertebral body height was obtained by dividing by VD. The vertebral body height (VH) calculation formula is shown as the following:

$$VH^i = \frac{1}{VD^i}\sum_{x=1}^{h}\sum_{y=1}^{w}P_{xy} \tag{9}$$

Among them, $h$ and $w$ respectively represent the height and width of the picture, $P_{xy}$ represents the pixel value when the height coordinate is $x$ and the width coordinate is $y$, and the value of $P_{xy}$ is 0 or 1.

*Disc height.* In the field of IVD height calculation, previous study showed that using area-based quantitative-measurement method was better than using point-based method, in which the result with excellent reliability showed that IVD height was equal to 60% or 80% of IVD diameter in sagittal view[14]. Therefore, in this study, the lumbar IVD height was calculated as 80% of lumbar-disc diameter.

After the feature-location points being obtained (Fig. 5i), the area of the lumbar IVD was calculated with the sum of all the pixel values between the two-line segments (Fig. 5j), while the lumbar IVD height was obtained by dividing by the lumbar IVD diameter. The IVD height (DH) calculation formula is shown as the following:

$$DH^i = \frac{1}{\mu||D_a^i D_p^i||^i}\sum_{x=\min X_D}^{\max X_D}\sum_{y=\min Y_D}^{\max Y_D}P_{xy} \tag{10}$$

Among them, μ represents the percentage of the center area of the entire lumbar IVD, taken as 80%, $||D_a^i D_p^i||^i$ represents the diameter of the $i$th lumbar IVD, and $X_D$ and $Y_D$ represent the width and height coordinate sets of the four characteristic location points respectively. $\{x_{D_{1a}^i}, x_{D_{2a}^i}, x_{D_{1p}^i}, x_{D_{2p}^i}\}$、 $\{y_{D_{1a}^i}, y_{D_{2a}^i}, y_{D_{1p}^i}, y_{D_{2p}^i}\}$

*Disc-height index.* To reduce individual differences, disc-height index (DHI) was used as normalized geometric parameter. Once the angle of the vertebral body and the midpoint of the endplate marked, the measurement line was drawn according to the marked point[15]. The DHI calculation formula is shown as the following:

$$DHI^i = \frac{2 \times DH^i}{VH^i + VH^{i+1}} \tag{11}$$

Among them, $DH^i$ represents the height of the $i$th lumbar IVD, and $VH^i$ and $VH^{i+1}$ respectively represent the height of the $i$th and the $(i+1)$th vertebral body.

*Disc height-to-diameter ratio.* Disc height-to-diameter ratio (HDR) is proposed to simultaneously characterize the height and shape of the IVD, which is considered to be the most accurate and repeatable[30]. In this study, the maximum IVD diameter was obtained by feature-location points, while average IVD height was calculated using the area-based method. Therefore, HDR calculation formula is

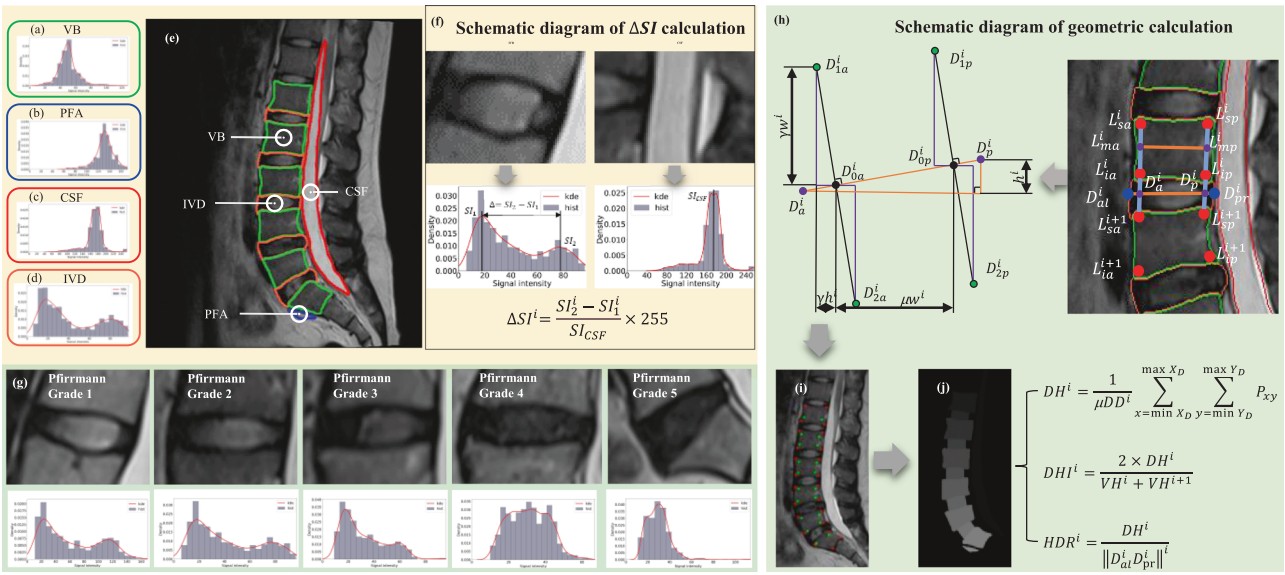

**Fig. 5 Scheme diagram of IVD parameter calculation.** Signal-intensity histogram calculation: (**a**) vertebral body area, (**b**) presacral fat area, (**c**) cerebrospinal fluid area, and (**d**) intervertebral disc area. **e** The outline of the segmented area is displayed on the original image. **f** Schematic diagram of $\Delta SI$ calculation. **g** Signal-intensity histogram corresponding to different Pfirrmann grade. **h** A geometric calculation method of lumbar disc-height parameters based on area, (**i**) vertebral body corner detection result (red points) and feature-point calculation result (green points), and (**j**) 80% area-extraction result of the intervertebral disc center.

shown as the following:

$$HDR^i = \frac{DH^i}{||D_{al}^i D_{pr}^i||^i} \tag{12}$$

where $||D_{al}^i D_{pr}^i||^i$ represents the maximum diameter of the $i$th lumbar intervertebral disc.

### IVD-degeneration quantitation

*Signal-intensity peak-deviation degree.* With IVD-degeneration process, water-content loss can be reflected in signal-intensity changes and height decrease. In this study, the signal-intensity peak-deviation degree from the center ($\Delta SI$) was mainly calculated to describe the water-content loss status with IVD degeneration. Based on IVDs with modified Pfirrmann grade (levels 1, 2, 3, 4, and 5–8), mean and standard deviation of the standard signal-intensity peak difference ($\Delta SI$) of each grade were established as grading standard to quantitatively analyze IVD degeneration (Supplementary Fig. 4). The calculation formula is shown as the following:

$$\triangle = \frac{||\triangle SI - \mu_{i+1}||}{\sigma_{i+1}} - \frac{||\triangle SI - \mu_i||}{\sigma_i} \tag{13}$$

where $\Delta SI$ is the current peak signal-intensity difference of the IVD, $\mu_i$ and $\sigma_i$ are the mean and standard deviation of the standard signal-intensity peak difference of the $i$th level, and $i$ is 1–4.

*Quantitative analysis on IVD degeneration.* For the original parameter DH, the ratio of the current DH to the average DH of the corresponding healthy intervertebral disc was used to calculate the collapse percentage. For the nonoriginal parameters $\Delta SI$, DHI, and HDR, which involve the influence of variables such as vertebral body height and intervertebral disc diameter, the same method as J.Jarman et al. is used to calculate the degree of deviation from the range center of the corresponding healthy intervertebral disc-height parameter[15]:

$$\beta_k = \frac{X - \mu_i^j}{\sigma_i^j} \tag{14}$$

where $\beta_k$ represents the degree of deviation between the $k$th nonoriginal parameter and the mean value of the corresponding healthy intervertebral-disc parameter, and k is from 1 to 3. When $\beta_k$ is smaller, the degree of signal intensity of the intervertebral disc degenerates or the collapse is higher. $j$ represents gender, which is 0 or 1, and $i$ represents structure position.

### Evaluation of model performance

*Accuracy evaluation on IVD-segmentation performance.* To ensure that the model trained with images from one hospital may present equally good accuracy in segmentation for all images from other three hospitals, 20 images randomly selected from each hospital were used to test.

Dice index and Intersection over Union (IOU) were used to measure the similarity between the segmented IVD-related areas and the manual labeled boundaries.

*Consistency evaluation on IVD-parameter quantitation.* In our study, accuracy in segmentation performance and consistency in IVD quantitative analysis are equally important. To evaluate consistency in IVD quantitative analysis in different resolution, 46 MR images with resolution of 320*320 were randomly selected from Data Set B to be segmented and quantified by model B. Meanwhile, these images were adjusted to 512*512 for segmentation and quantitation by model A. If IVD parameters extracted from model A and model B show good consistency, model A (trained with resolution of 512*512) will be considered applicable enough to extract IVD parameters among a larger population (Data Set C) with different machines.

Although manual measurement may present a greater error and lower consistency than machine measurement, IVD parameters measured by a senior radiologist and orthopedic residents are important as control standard. A 4th-year radiology resident (DW Kong), and a 4th-year orthopedic resident (MC Yin) measured and calculated all the IVD parameters (HDR and DHI) among these 15 MR images randomly selected from Data Set B. Each IVD was measured and recorded three times (Supplementary Fig. 1), from which mean values of three-time measurements were used to compare with each other. In addition, to avoid fatigue in long-term measurement, these residents were asked to have a 20-minute rest after measuring every two MR images.

The intraclass correlation coefficient (ICC) was used to analyze the consistency between the IVD-parameter extraction and IVD manual measurement. Mean time spent on each IVD quantitation was used to describe efficiency.

*Validity evaluation on IVD-degeneration quantitation.* To test the validity of signal-intensity quantitation on IVD degeneration, 46 MR images randomly selected from Data Set A and Data Set B, respectively, were used to automatically grade IVD-degeneration levels. Meanwhile, a research team, composed of a 4th-year radiology resident (DW Kong), two 8th-year orthopedic resident (J Chen, XF Ma), and three 4th-year orthopedic residents (YL Sun, YP Lin, and MC Yin), graded all the IVD-degeneration levels independently according to the modified Pfirrmann grading system[10]. They were all blinded to the automatic quantitative measures. Disagreements were resolved by consensus with additional two 10th-year orthopedic residents (XJ Cui and YJ Wang). MacroF1 score was used to analyze the validity between the automatic grade results and final manual grade results.

*Baseline characteristic of IVD parameters in a large population.* A retrospective study was conducted at four hospitals around China, in which the study population composed of patients who completed lumbar-spine MRI examination between January 1, 2019 and March 30, 2021. Further screening was conducted to exclude IVD herniation, lumbar spondylolisthesis, spine tumors, and severe ossification in vertebral bodies, whose abnormal signal-intensity distribution or irregular boundaries in IVD-related areas may enhance heterogeneity in IVD-degeneration

parameters. The screened MR images (Data Set C) were finally used to determine the relationships of baseline variables (age, gender, segments, and degeneration grades) and IVD quantitative parameters (ΔSI, DH, DHI, and HDR).

**Statistical analysis.** For performance evaluation in IVD segmentation and quantitation, the intraclass correlation coefficient (ICC) was used to analyze the consistency in IVD quantitation. The Dice coefficient and the Intersection over Union (IOU), also known as the Jaccard index, were used to evaluate the segmentation performance of the model. They are given by the following expression:

$$\text{mDice} = \frac{1}{C}\sum_{i=1}^{C}\frac{2*|G_i \cap P_i|}{|G_i|+|P_i|} \tag{15}$$

$$\text{mIOU} = \frac{1}{C}\sum_{i=1}^{C}\frac{|G_i \cap P_i|}{|G_i \cup P_i|} \tag{16}$$

where $G_i$ is the ground-truth annotation and $P_i$ is segmentation result for the *ith* segmentation area, and $C$ takes 14, indicating 14 segmented areas.

For IVD characteristic analysis, the mean and standard deviation were calculated for continuous variables and frequency and proportion for categorical variables. The following test was used: T-test, Mann–Whitney U for continuous variables, Chi-square for nominal variables, and Spearman rank correlation. MLR was carried out to determine the relationships of baseline variables (age, gender, segments, and degeneration grades) and IVD quantitative parameters (ΔSI, DH, DHI, and HDR). Spearman rank correlation analysis was used to investigate the correlation between IVD signal intensity and degeneration grades. The macroF1 score and the Kendall correlation coefficient were used to analyze the validity in IVD-degeneration grading performance.

An absolute value of r of 0–0.4 was considered as weak correlation, 0.4–0.6 as moderate correlation, and greater than 0.6 as strong correlation. p-value of <0.05 was considered statistically significant. The calculations were made using IBM SPSS Statistics (version 26, IBM, USA) and Stata (version 15.1, USA).

**Reporting summary.** Further information on research design is available in the Nature Research Reporting Summary linked to this article.

## Data availability
The raw demographic and MRI data are protected and are not publicly available due to hospital regulations, even all the identification has been removed. The data that support the findings of this study are available on request from the corresponding authors (YJ. Wang and YZ. Tian) for noncommercial, research purposes. Reply will be sent in two weeks.

## Code availability
Some of the core code generated or used during research is available in repositories or online: https://github.com/no-saint-no-angel/BianqueNet

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

## Acknowledgements

This study was supported by the National Key R&D Program of China (2020YFE0201600) and the National Natural Science Foundation of China (81930116, 81804115, 81873317, and 81730107).

## Author contributions

Guarantor of integrity of the entire study, Y.L.S. and Y.J.W.; study concepts/study design or data acquisition or data analysis/interpretation, all authors; paper drafting or paper revision for important intellectual content, all authors; approval of the final version of the submitted paper, all authors; agrees to ensure any questions related to the work are appropriately resolved, all authors; literature research, H.D.Z., M.C.Y., M.Y., and X.J.C.; clinical studies, Y.L.S., D.W.K., M.C.Y., J.C., Y.P.L., and X.F.M.; experimental studies, Y.Z.T., H.S.W., and G.J.Y.; statistical analysis, H.D.Z., M.Y., and X.J.C.; and paper revision, all authors.

## Competing interests

The authors declare no competing interests.
