## [Peer Review File · Nature Communications]

Reviewers' Comments:

Reviewer #1:

Remarks to the Author:

This paper proposes a semantic segmentation network (BianqueNet) to achieve high-precision segmentation of intervertebral disc (IVD) related areas and a quantitative method to calculate the signal intensity difference (ΔSI) in IVD, average disc height (DH), disc height index (DHI), and disc height-to-diameter ratio (DHR). The method in this paper is evaluated in a dataset contains 1051 MRI images collected from four hospitals around China. The merit of this paper is that its evaluation is implemented on quite a large dataset. However, the main drawback of this paper is it lacks methodology innovations. For example:

1. What is the advantage of the proposed method compared with the widely-used semantic segmentation method such as U-net?
2. What are the innovation parts in the proposed swin transform skip connection (ST-SC) module? Also, there are some issues in writing the paper:
3. Fig. 1 and 2 contain too much information, while the most important information is lost. For example, the proposed ST-SC module accounts for only a small proportion in Fig. 2; its modules, i.e., LN and W-MSA are not stated anywhere in the caption or in the text.
4. The paper does not describe how the signal intensity difference (ΔSI) in IVD, average disc height (DH), disc height index (DHI), and disc height-to-diameter ratio (DHR) are calculated after segmenting the intervertebral disc. Even if these calculation methods are not originally proposed by the authors, they should at least briefly describe how they are implemented.
5. In the experimental part, there seem to be no qualitative illustrations of the segmentation results. Also, the authors do not show how the segmentation results are used to analyze the correlation with IVD degeneration grading.

Reviewer #2:

Remarks to the Author:

This paper utilized CNN for lumbar spine segmentation to evaluate the intervertebral disc degeneration.

1. It is great to collect the dataset which may benefit the community. I'd like to ask if the dataset will be released or not?
2. The main contribution of this paper is utilizing a network for lumbar spine segmentation. The segmentation network showed in Fig.2, why do we need the ST-SC module? The multi-head self-attention mechanism can indeed model the long-range information dependency, while you have already used multi-scale information in the network, so I'm wondering if you indeed need the high-computational cost ST-SC module or not.
3. In Fig. 1, the pipeline shows that you input two datasets into the network, that's fine for data from these sites. However, then how do you guarantee your model works fine for data collected at a new site.
4. Is that really good to input different resolution images in the network? How about upsampling them to 512x512 for training and test on different resolution images? Why do we need multi-scale input in the training phase.
5. What's the significance of this work? The segmentation model (I don't think so) or the truth you proved that neural network can work well on evaluating the intervertebral disc degeneration?
6. What's the novelty of this work?
7. The main issue is that this kind of work (maybe the works are not about intervertebral disc degeneration but on other tasks, but they have proved the possibility of using networks for analyzing some diseases) has been vastly done before.

Reviewer #3:

Remarks to the Author:

Thank you for the opportunity to review this manuscript. The broad goal of developing accurate quantitative measures of IVD degeneration using deep learning is important in my opinion. Without these measures it will not be possible to work towards differentiating degeneration due to aging compared to clinically important degeneration. This is a major gap in the back pain field. I have very limited understanding of the deep learning methods described in this paper, so am unable to comment on the quality of many the methods undertaken. My comments are limited to other aspects of the manuscript.

1. The aim is unclear to me and does not capture the big picture of what this work could achieve. While the long-term potential clinical use is important but I am not sure the current study has this aim/purpose.

2. Abstract refers to reliability with Pfirrmann. I think correlation is a better term to describe the study.

3. In the Implications for patient care I believe the authors have extrapolated beyond their study results too far. For example, they should not talk about assessing risk of disc herniation. This is inappropriate based on current study. In general the conclusions need to be more limited the study design and results. This study lays a foundation to later test the clinical importance of MRI measures of disc degeneration, but does not investigate this clinical aspect. This should be clearer throughout.

4. There are a reasonable number of small expression issues.

5. I cannot assess if the methods to test if the approach works well across different MRI machines is valid. A radiologist review of this would be important.

6. The methods do not provide adequate information on the process for assessing Pfirrmann scores. There is also no reference for the modified scale. The reader needs to understand this scale to interpret much of the findings. Was reliability of Pfirrmann assessed. Line 126 reports "discussed together". This is unclear. Were those doing Pfirrmann measures blinded to the quantitative measures?

7. It does not seem that any normalisation was performed for disc height. Interpreting the raw score requires this to make it cleanly useful. The nice graphs in figure 4 suggest that individuals can be compared to normal values, means etc based on age, disc level etc. More discussion of this would be good as this demonstrates how the findings could be applied in clinical practice and assessed for ability to predict important outcomes in future studies.

8. How many images were manually segmented to compare and train the automatic segmentation. As mentioned, I have little expertise in this area but it would be good if the processes could be made easier to read for an average reader especially clinical experts who will want to understand this work.

9. What exactly is "signal intensity difference". Why is it not just signal intensity? IS this value the mean SI in the region of interest or max value etc. Given how critical the signal intensity difference and the 3 measures of DH are to the paper, they should be more clearly defined and explained.

10. Line 244 describes nearly 1/3 of potential participants being excluded. It seems this would limit the generalisability of the results.

11. It seems odd that DH increases with age (line 258). Please explain or discuss.

12. More clearly state reference groups for analyses in table 5

13. Many values (e.g those in table 5) are described based on statistical significance. It would be helpful to understand how much some of the variables influences the outcome (e.g disc height). Can values like R2 be provided. So how much is the difference based on gender for example. Is it likely to be important?

14. Please define BMP

15. Third limitations is unclear line 329

16. Figure 3 described Pfirrmann in grade 1-5 which does not appear to match the modified version used.

Reply to editors and reviewers

Hello Dear editors and reviewers,

Thank you for your kind suggestions and comments. I've revised this manuscript to address all of the reviewer comments.

My point-by-point reply is as follows:

Reviewers' comments:

Reviewer #1:

Manuscript Summary:

This paper proposes a semantic segmentation network (BianqueNet) to achieve high-precision segmentation of intervertebral disc (IVD) related areas and a quantitative method to calculate the signal intensity difference (ΔSI) in IVD, average disc height (DH), disc height index (DHI), and disc height-to-diameter ratio (DHR). The method in this paper is evaluated in a dataset contains 1051 MRI images collected from four hospitals around China. The merit of this paper is that its evaluation is implemented on quite a large dataset. However, the main drawback of this paper is it lacks methodology innovations. For example:

1. What is the advantage of the proposed method compared with the widely-used semantic segmentation method such as U-net?

Reply to Reviewer: This point is well taken. Please forgive us for not making this point clearer in the first submission. The most important advantage of the proposed network is higher accuracy in segmentation. As shown in the figure and table below, compared with U-Net, higher segmentation may contribute better feature extraction, which is important in IVD degeneration quantitation.

Performance of BianqueNet compared with other semantic segmentation method

model	Vertebral body		IVD		Lumbar spine	
	mDice	mIoU	mDice	mIoU	mDice	mIoU
U-Net	0.9159	0.8732	0.8886	0.8260	0.9020	0.8460
PSPnet	0.9334	0.8849	0.9066	0.8389	0.9135	0.8520
BianqueNet	0.9703	0.9425	0.9480	0.9019	0.9470	0.9035

Figure 2. The segmentation performance of BianqueNet in three typical cases and the influence of different segmentation accuracy on feature point detection and calculation.

Fig.2 in the revised paper shows the importance of segmentation accuracy.

Similar to U-NET, BianqueNet is an encoder-decoder network architecture.

In the encoder section, we adopted the method of Deeplabv3 +, feature extraction network (applying empty convolution in deep separable) and added a DFE module (extracting richer global semantic information).

In the decoder section, compared with U-Net, we proposed ST-SC module (extracting more accurate contour details) to replace the original Skip Connection Section to obtain the final denser prediction results by means of integrating the feature map information of each stage from up-sampling process.

2. What are the innovation parts in the proposed Swin-Transform skip connection (ST-SC) module?

Reply to Reviewer: In the revised paper, we described the innovation parts in detail (Line 437-462).

To obtain more accurate segmentation information of IVDs and vertebral bodies, down-sampling-origin feature maps information needed to be selectively transferred to

the up-sampling paths. Swin-Transformer blocks containing multi-headed self-attention mechanisms were added to the Skip Connection (see text for more details on the structure) to achieve the selective transfer function more efficiently and with less complexity.

To verify this idea, we output feature maps from both network with and without ST-SC module as shown in **Fig.5 f, g**.

Feature maps in the upper row are output from network with ST-SC module, and those in the bottom row are output from network with 1×1 convolution.

For high-resolution and low-path feature maps (left column), there are not significantly difference in the visualization between these two networks.

However, for middle-path feature maps (middle column), boundary information of vertebral bodies and cerebrospinal fluid presents clearer from the network with ST-SC module.

For low-path feature maps (right column), boundary information of IVDs and cerebrospinal fluid presents clearer from the network with ST-SC module.

Notably, this method may be the first one to be introduced into spine MR image segmentation tasks. In general, the innovation of our method is reflected in the compatibility and accuracy improvement of application in lumbar MR images quantitation.

Also, there are some issues in writing the paper:

3. Fig. 1 and 2 contain too much information, while the most important information is lost. For example, the proposed ST-SC module accounts for only a small proportion in Fig. 2; its modules, i.e., LN and W-MSA are not stated anywhere in the caption or in the text.

Reply to Reviewer: Thank you for pointing out these issues. Considering that there is too much information in Fig.1 and Fig.2, we redrew **Fig.1** with a clearer study design and workflow, as shown below.

Figure 1. Study design and clinical workflow.

Meanwhile, we separated **Fig.2** into two separate figures (**Fig.5** for network architecture, **Fig.6** for feature extraction calculation).

Figure 5. The proposed Bianque-net consists of segmentation CNNs with DFE module and SW-SC module.

Figure 6. Schematic diagram of IVD parameters calculation.

In addition, further explanations were added in the revised manuscript (**Line 407-558**).

4. The paper does not describe how the signal intensity difference (ΔSI) in IVD, average disc height (DH), disc height index (DHI), and disc height-to-diameter ratio (DHR) are calculated after segmenting the intervertebral disc. Even if these calculation methods are not originally proposed by the authors, they should at least briefly describe how they are implemented.

Reply to Reviewer: Thank you for your comment. A long and detailed calculation description for every parameter has been provided in the Supplemental file, while there was less information of the detailed method in the main text, which may have confused the reader as to the important parts.

Thus, in the revised paper, we put these parts briefly into the main text (**Line 484-558**) and used **Fig.6** to help readers understand the details.

5. In the experimental part, there seem to be no qualitative illustrations of the segmentation results. Also, the authors do not show how the segmentation results are used to analyze the correlation with IVD degeneration grading.

Reply to Reviewer: We only calculated segmentation performance of different networks in the previous submission, which may have failed to make clear the segmentation performance differences.

In the revised paper, we added **Fig.2** to show vital differences in segmentation between BianqueNet and U-Net network, and made a certain analysis of the role of

each module (**Table 1**).

Based on segmentation, we extracted the signal intensity and geometric characteristics of the IVDs (see **Methods/Lumbar IVD quantitative analysis/ Parameters Calculation based on IVD-related area segmentation** for details).

On the other hand, all IVDs were manually graded according to the modified Pfirrmann grading guidelines.

For the correlation analysis of ΔSI and degeneration grade, considering that there is no difference in signal intensity values of grades 5-8, we combined grades 5-8 into one class and conducted a correlation analysis test for disc ΔSI parameters and corresponding degeneration grade (1, 2, 3, 4, 5-8).

For the correlation analysis of geometric parameters and degenerative grade, we classified grade 1-5 into one class, given that the IVD height between grade 1 and grade 5 are basically equal (no collapse occurs).

Fig.7 was added to the revised paper to present the different geometric characteristics and signal intensity among all kinds of IVD degeneration grades.

Figure 7. Schematic of IVD degeneration quantitation.

Reviewer #2:

Manuscript Summary:

This paper utilized CNN for lumbar spine segmentation to evaluate the intervertebral disc degeneration.

1. It is great to collect the dataset which may benefit the community. I'd like to ask if the dataset will be released or not?

Reply to Reviewer: We do realize that if this dataset is released publicly, it may be used by others to perform important research. However, for ethical reasons its release has to be limited. Even so, publication of the current study should help popularize this efficient and consistent method among the research community to help establish a consistently growing and updated dataset with proper privacy protections in place.

2. The main contribution of this paper is utilizing a network for lumbar spine segmentation. The segmentation network showed in Fig.2, why do we need the ST-SC module? The multi-head self-attention mechanism can indeed model the long-range information dependency, while you have already used multi-scale information in the network, so I'm wondering if you indeed need the high-computational cost ST-SC module or not.

Reply to Reviewer: We appreciate the comment. Please forgive us as the details and novelty of our segmentation network were not described clearly in the previous manuscript, which has now been clarified in the revised paper (**Line 407-483**).

As one of the most important parts of our work, segmentation provides a basis for quantitative features automatic extraction. As we know, the feature maps from depth feature extraction module contain rich multi-scale semantic information and little boundary detail information. Considering that deconvolution operation presents limited capacity in recovering detail information from target image, and the use of skip connection can better improve this problem. Furthermore, the multi-head self-attention mechanism in the skip connection module (SC module) can selectively transfer the down-sampling feature map information to the up-sampling path, providing more accurate details of target images for the up-sampled path.

To verify this idea, we output feature maps from both networks, with and without an ST-SC module, as shown in **Fig. 5f, g**.

Feature maps in the upper row are output from network with ST-SC module, and those in the bottom row are output from network with 1*1 convolution.

For high-resolution and low-path feature maps (left column), there are no significant differences in the visualization between these two networks.

However, for middle-path feature maps (middle column), boundary information of vertebral bodies and cerebrospinal fluid is presented clearer from the network with ST-SC module.

Likewise, for low-path feature maps (right column), boundary information of IVDs and cerebrospinal fluid is presented clearer from the network with ST-SC module.

We also compared final segmentation outputs and their related feature points in **Fig.2**.

Case1 and Case2 show that our network may predict the boundary information of IVDs and vertebral bodies better.

On the other hand, compared with global self-attention mechanism, computational complexity of ST-SC module is lower. As a major contributor for computational complexity in ST-SC module, W-MSA (SW-MSA) calculates self-attention based on local windows, which are arranged to evenly partition the image in a non-overlapping manner. Supposing each window contains $M \times M$ patches, the computational complexity of a global MSA module and a window based one on an image of hw patches are:

$$\Omega(MSA) = 4hwC^2 + 2(hw)^2C$$

$$\Omega(W-MSA) = 4hwC^2 + 2M^2hwC$$

where the global MSA module is quadratic to patch number hw , and the latter is linear when M is fixed. Global self-attention computation is generally unaffordable for a large hw , while the window based self-attention is scalable.

3.In **Fig. 1**, the pipeline shows that you input two datasets into the network, that's

fine for data from these sites. However, then how do you guarantee your model works fine for data collected a new site.

Reply to Reviewer: The pipeline in the previous manuscript was not clear enough. To better explain our study design and workflow, we redrew the pipeline in Fig. 1

As shown in Fig.1 above, both Data Set A (resolution of 512*512) and Data Set B (resolution of 320*320) are from two machines in the Longhua hospital, Shanghai University of TCM. Model A and Model B were trained with images at different resolutions.

Considering that the resolutions of MR images from different hospitals are not consistent (320*320, 512*512, 640*640, 960*960, etc), we proposed to adjust all the MR images to a uniform resolution of 512*512 to improve model’s universality. Furthermore, we carried out a validation test to determine whether the adjustment of MR images resolution may affect the accuracy of IVD degeneration quantitation or not, whose results were described in the section (Quantitation performance in different MR images with different resolutions, Line 149-168).

The results show that Model A could still accurately calculate IVD degeneration parameters after MR images resolution were adjusted to 512*512 (Table 3).

Table 3 Consistency analysis of intervertebral disc parameters calculated by MRI of different sizes

Measure	Intraclass Correlation ^b	
	ICC ^a	95% CI
ΔSI	.874***	(.840, .902)
DHI	.958***	(.943, .968)
HDR	.956***	(.886, .978)

Therefore, the subsequent characteristic parameters of IVD degeneration were segmented and calculated by Model A.

The reason why we set MR images resolution to 512*512 for final model input is because a large proportion of images present resolution of 512*512 among all collected MR images. Considering that using an interpolation method may miss or change information from an image by reducing or enlarging images sizes, we finally chose the middle image resolution of 512*512 to retain the original information of MR images to the most extent.

To ensure that this network can perform well in other data sets, unrestricted histogram equalization operation and data enhancement operation were used to improve the generalization performance of the model in the training phase.

In addition, MR images were randomly selected from the other three hospitals, adjusted resolution to 512*512, and evaluated consistency with Model A, as shown in **Table 2**.

Table 2. Segmentation Performance on other MR images from different sites

Cites	Vertebral body		IVD		Lumbar spine	
	mDice	mIoU	mDice	mIoU	mDice	mIoU
Dongzhimen Hospital, Beijing University of CM	0.9567	0.9214	0.9198	0.8567	0.9193	0.8620
Guangdong Provincial Hospital of CM	0.9656	0.9498	0.9337	0.9046	0.9365	0.8888
Shenzhen Pingle Orthopedics Hospital	0.9654	0.9445	0.9334	0.8952	0.9269	0.8763
Longhua Hospital, Shanghai University of TCM *	0.9703	0.9425	0.9480	0.9019	0.9470	0.9035

* MR images from Longhua Hospital, Shanghai University of TCM were used to train the model as data set and to evaluate the accuracy of segmentation performance as control.

The segmentation performance for MR image from Dongzhimen Hospital was

acceptably moderate, while those for other two hospitals showed no significant difference with training set (Longhua Hospital).

4. Is that really good to input different resolution images in the network? How about upsampling them to 512x512 for training and test on different resolution images? Why do we need multi-scale input in the training phase?

Reply to Reviewer: Thank you for your comment. We apologize for our mistake in the previous pipeline. We have revised them in the current manuscript.

Please see our explanation above as to the reason why we set MR images resolution of 512*512 for the final model input.

5. What's the significance of this work? The segmentation model (I don't think so) or the truth you proved that neural network can work well on evaluating the intervertebral disc degeneration?

Reply to Reviewer: I'm sorry for the misunderstanding.

The significance of this work is to propose a segmentation-based and automatic quantitative analysis method for IVD parameters. To achieve quantitative evaluation of IVD degeneration, MR images of large sample were collected from several hospitals to extract parameters and establish baseline characteristics of lumbar IVDs in different subgroups (age, gender, segment, and Pfirrmann grade) as IVD degeneration grading criterion. The segmentation network or neural network were only used to obtain the geometric feature information of IVD-related areas as the input for the subsequent IVD features extraction. According to grading criterion, we achieved automatic grade IVD degeneration with quantitative features of IVDs as **Fig.4**.

Figure 4. Quantitative analysis results of typical cases.

6. What's the novelty of this work?

Reply to Reviewer: The innovations can be summarized in three points:

- 1) We propose an improved Deeplabv3 + semantic segmentation network.
- 2) Based on accurate and consistent segmentation, we propose an automatic quantitative method in IVD degeneration feature extraction.
- 3) We establish a baseline characteristic of IVD in different subgroups (age, gender, segments and degeneration grades) among a large population.

7. The main issue is that this kind of work (maybe the works are not about intervertebral disc degeneration but on other tasks, but they have proved the possibility of using networks for analyzing some diseases) has been vastly done before.

Reply to Reviewer: We apologize for not making this point clearer in the previous version of the paper. Previous CNN were mainly proposed as a degeneration level classifier, while our network is proposed to automatically quantify IVD degeneration, which is the major difference from other medical deep learning-based networks.

Indeed, many studies were focused on IVD degeneration. Some quantitative studies proposed some semi-automatic quantitative analysis with general medical image processing software, while other deep learning studies proposed some classifiers for automatic IVD degeneration grading, which were similar to some auxiliary diagnostic neural networks, such as the segmentation of brain tumors and the classification of tumor types, etc.

Compared with other quantitative analysis studies on IVD degeneration, our work can achieve both automatic extraction of IVD degeneration parameters and quantitative analysis IVD degeneration.

Compared with other CNN studies on auxiliary diagnosis, our work embodies not only the deep learning neural network approach, but also a good correlation between characteristics of IVD parameters and IVD degeneration grades, which may improve the interpretability of diagnostic results.

In terms of method, different from most methods in other studies, segmentation network is only a part to achieve automatic IVD feature extraction. Subsequently, an automatic quantitative feature calculation method is proposed to comprehensively analyze the geometric characteristic (DH, DHI, HDR) and signal intensity (ΔSI) of IVD based on segmentation, which may quantitatively reflect structural collapse and water content loss with the process of IVD degeneration.

Reviewer #3:

Manuscript Summary:

Thank you for the opportunity to review this manuscript. The broad goal of developing accurate quantitative measures of IVD degeneration using deep learning is important in my opinion. Without these measures it will not be possible to work towards differentiating degeneration due to aging compared to clinically important degeneration. This is a major gap in the back pain field. I have very limited understanding of the deep learning methods described in this paper, so am unable to comment on the quality of many the methods undertaken. My comments are limited to other aspects of the manuscript.

1. The aim is unclear to me and does not capture the big picture of what this work could achieve. While the long-term potential clinical use is important but I am not sure the current study has this aim/purpose.

Reply to Reviewer: Thank you for pointing out this issue. We didn't clearly state our research aim and main function of our network in the previous manuscript. We have now described all of the study design in detail (Line 374-398) and redrew Fig.1 to better illustrate our workflow.

In this study, we propose an improved deeplabv3+ segmentation network with newly designed modules and a quantitative method to automatically quantify IVD degeneration.

Firstly, we test its performance in automatic segmentation (accuracy) and calculation (consistency)

Secondly, we test its adaptability (segmentation and quantitation) from other machine in different hospitals.

Thirdly, IVD parameters were extracted from lumbar MR images by proposed network. According to large population subgroups (age, gender, segment, degeneration grade), we established a baseline criterion of IVD parameters characteristic for IVD degeneration automatic classification.

Figure 4. Quantitative analysis results of typical cases.

This study proposed a consistent, accurate and efficient deep learning network to achieve automatic lumbar IVD degeneration quantitation. We believe that the popularization of this method will greatly transfer MR images into data to establish a quantitative index system.

2.Abstract refers to reliability with Pfirrmann. I think correlation is a better term to describe the study.

Reply to Reviewer: Thank you for pointing this out. We used the wrong expression in the sentence ‘while signal intensity in IVD degeneration had excellent reliability according to the modified Pfirrmann Grade (macroF1=90.63%~92.02%)’.

MacroF1 value was used to evaluate the classification accuracy of proposed model. What we originally meant to express is that our method is very accurate in classification, but it is admittedly improper to express its use as ‘reliable’.

3.In the Implications for patient care I believe the authors have extrapolated beyond their study results too far. For example, they should not talk about assessing risk of disc herniation. This is inappropriate based on current study. In general the conclusions need to be more limited the study design and results. This study lays a foundation to later test the clinical importance of MRI measures of disc degermation but does not investigate this clinical aspect. This should be clearer throughout.

Reply to Reviewer: Thank you for your suggestion. In the previous version, we did extrapolate the application of our network.

As we mentioned in the *Introduction*, the progressive process of IVD degeneration may hardly be described by traditional degeneration grade. As an early phase of spine degenerative disease, IVD degeneration should be screened with more accurate data extracted from MRIs to prevent progressive outcomes. Although manual measurement on IVD is convenient to learn and apply, it may take several attempts and a high degree of concentration to ensure accuracy in its measurements. As a result of these man-power issues, data from spine MRIs haven’t been fully and widely used in some clinical practices and clinical trials with large populations. Due to the nature of its automation our method may provide more precise information for clinical practice (lumbar MR image structural report), clinical trials (efficacy assessment) and mechanism investigation (biomechanics research and finite element analysis).

4.There are a reasonable number of small expression issues.

Reply to Reviewer: Thank you. We have carefully checked and improved the English

grammar and syntax in the revised manuscript.

5.I cannot assess if the methods to test if the approach works well across different MRI machines is valid. A radiologist review of this would be important.

Reply to Reviewer: Thank you for your comment and agree with your concern. To test the model performance, especially in clinical settings, we designed three different evaluations for a human-machine comparison, as described in the *Methods* (Line 588-628).

∨ Evaluation of model performance

Accuracy evaluation on IVD segmentation performance

Consistency evaluation on IVD parameter quantitation

Validity evaluation on IVD degeneration quantitation

In the segmentation phase, we compared manual segmentation with model segmentation. mDice and mIoU were used to assess the similarity between regions segmented by machines and regions defined by residents.

In the quantitation phase, we compared manual measurements with model calculation. IVD parameters measured by a senior radiologist and orthopedic residents are important as a control standard. A 4th-year radiology resident, and a 4th-year orthopedic resident measured and calculated all the IVD parameters (HDR and DHI) among these 15 MR images randomly selected from data set B.

Each IVD was measured and recorded three times, from which mean values of three-time measurements were used to compare with each other. In additions, to avoid fatigue in long-term measurement, these residents were asked to take a 20-minute rest after measuring every two MR images.

Supplement Fig.2 Schematic diagram of manual measurement of IVD

The intraclass correlation coefficient (ICC) was used to analyze the consistency between the IVD parameters extraction and IVD manual measurement. Results were reported in the revised manuscript (**Line 169-196**).

Finally, in the IVD degeneration classification phase, to test the validity of signal intensity quantitation on IVD degeneration, 50 MR images randomly selected from data set A and data set B, respectively, were used to automatically grade IVD degeneration levels. Meanwhile, a research team, composed of a 4th-year radiology resident, two 8th-year orthopedic residents and three 4th-year orthopedic residents, graded all the IVD degeneration levels independently, according to the modified Pfirrmann Grading System. They were all blinded to the automatic quantitative measures. Disagreements were resolved by consensus with additional two 10th-year orthopedic residents.

MacroF1-score and Kendall coefficient were used to analyze the validity between the automatic grade results and final manual grade results. Results were reported in the revised manuscript (**Line 222-253**).

6. The methods do not provide adequate information on the process for assessing Pfirrmann scores. There is also no reference for the modified scale. The reader needs to understand this scale to interpret much of the findings. Was reliability of Pfirrmann assessed. Line 126 reports “discussed together”. This is unclear. Were those doing Pfirrmann measures blinded to the quantitative measures?

Reply to Reviewer: We apologize for this issue as some important details were omitted in the previous manuscript, but we have added this information in the revised paper.

Regarding the modified Pfirrmann Grade, we inserted a **related reference** and included an illustration of different grades of IVD degeneration in **Fig.7**, as shown below (lower right part)

Figure 7. Scheme of IVD degeneration quantitation.

To ensure the reliability of Pfirrmann grading, a research team, composed of a 4th-year radiology resident (DW Kong), two 8th-year orthopedic resident (J Chen, XF Ma), and three 4th-year orthopedic resident (YL Sun, YP Lin, MC Yin), graded all the IVD degeneration levels independently according to the modified Pfirrmann Grading System. They were all blinded to the automatic quantitative measures. Disagreements were resolved by consensus with additional two 10th-year orthopedic residents (XJ Cui and YJ Wang). In the revised version, we added more details about “discuss together” (Line 623-628).

7. It does not seem that any normalisation was performed for disc height. Interpreting the raw score requires this to make it cleanly useful. The nice graphs in figure 4 suggest that individuals can be compared to normal values, means etc based on age, disc level etc. More discussion of this would be good as this demonstrates how the findings could be applied in clinical practice and assessed for ability to predict important outcomes in future studies.

Reply to Reviewer: Thank you very much for your suggestion. Of course, there is no way to compare our IVD height (DH) with other measured DH in previous studies, because our DH is calculated with the number of pixels in specific segments, which depends on the MRI resolution (The larger the resolution of the image, the more pixel values, the larger the calculated DH).

To ensure the comparability of DH, the only thing we did for normalization was to adjust all the images to a fixed resolution (512*512), so that the number of pixels per unit area was kept consistent.

However, DH value calculated by our model is still an intermediate variable, which may be influenced by a number of other individual factors. According to the application of these two parameters (DHI, HDR) in previous studies, we calculated them with heights and diameters of IVD and VB for normalization and comparison.

Regarding future clinical practice and assessment, we will insert this network into MR image system and export a structural lumbar MRI report like Fig.4 for doctors, patients, and researchers. Compared with traditional text description MRI report, our quantitative report may provide more accurate IVD parameters to reflect height collapse and water content loss with IVD degeneration. According to IVD baseline characteristic criteria in each age, gender, and segments, deviation of IVD geometric parameters and pfirrmann grade based on signal intensity will be obtain automatically to reflect both structural collapse status and water content loss in IVD comprehensively, which may provide more precise information for clinical practice (lumbar MR image structural report), clinical trials (efficacy assessment) and mechanism investigation (biomechanics research and finite element analysis). Notably, this baseline characteristics will be updated dynamically as these MR image data are collected and summarized.

8. How many images were manually segmented to compare and train the automatic segmentation. As mentioned, I have little expertise in this area but it would be good if the processes could be made easier to read for an average reader especially clinical experts who will want to understand this work.

Reply to Reviewer: Thank you very much for this good suggestion. We have revised the manuscript to include this information.

In this study, we trained two models for segmentation using MRIs with different resolutions. The data set for Model A contains 223 participants' MRIs, including 303 images in the training set and 80 images in the test set. The data set for Model B contains 63 participants' MRIs, including 93 images in the training set and 24 images in the test set. The training set and test set were randomly assigned. The training images of two data sets were enhanced with data enhancement. **(Specific values are also shown in Fig.1)**

9. What exactly is “signal intensity difference”. Why is it not just signal intensity? IS this value the mean SI in the region of interest or max value etc. Given how critical the signal intensity difference and the 3 measures of DH are to the paper, they should be more clearly defined and explained.

Reply to Reviewer: Thank you for your comment. In the previous submission, there's a long and detailed calculation description for every parameter uploaded as Supplemental file, while there's less information of the detailed methodology in the main text, which we understand now may confuse the reader. In the revised paper, we outlined these parts briefly in the main text **(Line 484-558)** and used **Fig.6** to illustrate the calculation process.

10. Line 244 describes nearly 1/3 of potential participants being excluded. It seems this would limit the generalizability of the results.

Reply to Reviewer: We appreciate this concern. This does limit the versatility of this network to a certain extent. However, among the 1508 MRIs screened in 4 sites around China, there're 73 excluded for unaligned outlines (diagnosed as lumbar spondylolisthesis), 45 excluded for abnormal signal intensity distribution (diagnosed as spine tumors), 364 excluded for irregular structures (diagnosed as IVD herniation or vertebral body ossification), and only 144 excluded for imaging quality (segmentation results and corner detection did not meet the requirements of parameter calculation), and finally a total of 1051 individuals were collected **(Line 198-203)**.

Even though BianqueNet's performance is very good, future work will continue to optimize segmentation performance. There are generally two strategies to improve segmentation performance:

- 1) To improve the generalization performance of segmentation algorithm, one of the more reliable methods is to increase training samples, but this costs a lot of time.
- 2) To study more powerful segmentation algorithms. These measures may broaden the scope of our approach.

11. It seems odd that DH increases with age (line 258). Please explain or discuss.

Reply to Reviewer: Thank you very much for your concern. According to our statistical results, this is indeed the trend, which is similar with the age trend of peak bone mass.

According to the figure⁴, turning point for peak bone mass is age of 36. In our study, we found that the turning point for ‘peak IVD height’ is in age range of 50-60, which may be a secondary degenerative process. Due to changes in vertebral osteoporosis, the endplate of the vertebral body becomes more depressed, which may make the IVD sink into the vertebral body, resulting in lower vertebral height and higher disc height^{1,3}.

Also, there are many studies on the relationship between age and height of IVD and vertebral bodies. H.S. Monoo-Kuofi et al² concluded that IVD height increases with age, but not in a linear fashion, with alternating periods of overgrowth and thinning, and a significant decrease of 2.5% after age 50. These studies support our results.

In the future, we will continue to collect data from more MRIs and possibly investigate these IVD parameters as a function of aging.

12. More clearly state reference groups for analyses in table 5

Reply to Reviewer: Thank you for your suggestion.

This experiment is to investigate the influence of age, gender, and segments on these IVD parameters in degeneration progress. For each factor, when a reference group was selected in the Stata software, group of other factors was analyzed as controls. For example, when male factor is selected as the reference group, all data with female are included into the control group.

For age, 20-30 was selected as the reference group.

For segment, L4-L5 was selected as the reference group.

13.Many values (e.g those in table 5) are described based on statistical significance. It would be helpful to understand how much some of the variables influences the outcome (e.g disc height). Can values like R2 be provided. So how much is the difference based on gender for example. Is it likely to be important?

Reply to Reviewer: Our R^2 results showed that for the 4 IVD parameters, our model and actual values have a high goodness of fit.

	ΔSI	DH	DHI	HDR
Prob>F	0.0000	0.0000	0.0000	0.0000
R-squared	0.4226	0.4770	0.3909	0.3078
Adj R-squared	0.4191	0.4739	0.3872	0.3036
Root MSE	22.253	1.7014	0.04576	0.02744

Here, multiple regression analysis was standardized to investigate the influence of different factors (age, gender, segments) on IVD degeneration (ΔSI , DH, DHI, HDR). The greater absolute value of the normalized coefficient (only significant regression coefficient was concerned), the higher these dependent factors may influence on IVD degeneration.

For example, gender and other factors have no significant influence on ΔSI . For DH parameters, segment and gender had greater influence than age.

14.Please define BMP

Reply to Reviewer: BMP, short for Bitmap, is a standard image file format in The Windows operating system. It is supported by various Windows applications and widely used with the popularity of Windows operating system and the development of rich Windows applications.

15.Third limitations is unclear line 329

Reply to Reviewer: Thank you for your suggestion. What we mean is that this retrospective study may be subject to potential selection bias. We have revised all the limitations as below:

Our study has some limitations. First, this retrospective study may be subject to potential selection bias. Some prospective studies should be rigorously conducted to test the clinical utility of this proposed model. Second, our deep learning model was

trained and tested using a single ethnic group (namely, Chinese patients), so its reproducibility among different ethnic groups should be further evaluated.

In the future, it will be important to combine radiomics and prospective design and integrate all kinds of clinical examination, fluid flow biomechanics, and molecular approaches to improve accuracy in IVD degeneration evaluation.

16. Figure 3 described Pfirrmann in grade 1-5 which does not appear to match the modified version used.

Reply to Reviewer: According to the Modified Pfirrmann grading guidelines, the characteristics of IVD signal intensity are the same among grade 5 to 8, so IVD degeneration grade of 5 to 8 are combined as one (grade 5).

We inserted a related reference and drew an illustration of different grades of IVD degeneration in Fig.7, as shown above (lower right part)

References

1. Berlemann, U., Gries, N. C. & Moore, R. J. The relationship between height, shape and histological changes in early degeneration of the lower lumbar discs. *Eur. Spine J.* **7**, 212–217 (1998).

2. Amonoo-Kuofi, H. S. Morphometric changes in the heights and anteroposterior diameters of the lumbar intervertebral discs with age. *J. Anat.* **175**, 159–68 (1991).
3. Twomey, L. & Taylor, J. Age changes in lumbar intervertebral discs. *Acta Orthop.* **56**, 496–499 (1985).
4. Zhu X, Zheng H. Factors influencing peak bone mass gain. *Front Med.* 2021 Feb;15(1):53-69.

Reviewers' Comments:

Reviewer #1:

Remarks to the Author:

My questions are answered.

Reviewer #3:

Remarks to the Author:

The authors have responded to my previous comments and suggestions adequately. Responses were detailed but the changes were not highlighted so difficult to check if and when the additional information was added to the manuscript. A few minor ongoing comments:

- regarding my previous comment 13. The authors state the values are standardised values. This is not clear when I read the table. Please add this as a footnote etc.

- Regarding my previous comment 9. The authors have added additional descriptions as requested but I must say I still don't understand the description of signal intensity. Why is the SI measure a difference between 2 peaks? This is most likely my lack of expertise in the methods used in the paper; however, it does raise the potential issue that many readers will find this paper too dense to understand and this may detract from the potential value of the study. Please consider trying to simplify some of the reporting for people who are not experts in the methods but may be the researchers or clinicians who apply these findings in the future.

Reply to editors and reviewers

Hello Dear editors and reviewers,

Thank you for your kind suggestions and comments. I've revised this manuscript to address all of the reviewer comments.

My point-by-point reply is as follows:

Reviewer #3' comments:

The authors have responded to my previous comments and suggestions adequately. Responses were detailed but the changes were not highlighted so difficult to check if and when the additional information was added to the manuscript. A few minor ongoing comments:

Reply to Reviewer: Thank you for your kind suggestion. Please forgive us for not highlighting changes clearer in the revised version, because it has been extensively rewritten with outline rearrangement.

1. Regarding my previous comment 13. The authors state the values are standardised values. This is not clear when I read the table. Please add this as a footnote etc.

Reply to Reviewer: This point is well taken. Thanks. In the latest version, a footnote has been added in the Table 4 as below:

Table 4 Correlations between IVD parameters and modified Pfirrmann Grading

lumbar level	ΔSI	DH ^a		DHI		HDR	
		female	male	female	male	female	male
L1/L2		-.421***	-.296***	-.304***	-.235***	-.473***	-.397***
L2/L3	-.966***	-.481***	-.417***	-.354***	-.398***	-.575***	-.455***
L3/L4		-.639***	-.470***	-.530***	-.443***	-.626***	-.539***
L4/L5		-.656***	-.696***	-.560***	-.665***	-.709***	-.758***
L5/S1		-.701***	-.687***	-.641***	-.664***	-.744***	-.778***

*** p<0.01 ** p<0.05 * p<0.1

r, Spearman rank correlation coefficients

a, DH is the only parameter that is not standardized, while ΔSI can be applied to MRI at different centers, and DHI and HDR can be applied to different types of imaging means and physical measurements.

2. Regarding my previous comment 9. The authors have added additional descriptions as requested but I must say I still don't understand the description of signal intensity. Why is the SI measure a difference between 2 peaks? This is most likely my lack of expertise in the methods used in the paper; however, it does raise the potential issue that many readers will find this paper too dense to understand and this may detract from the potential value of the study. Please consider trying to simply some of the reporting for people who are not experts in the methods but may be the researchers or clinicians who apply these findings in the future.

Reply to Reviewer: Thank you for your kind suggestion. It is very important to simply some of the reporting for people, especially for popularization of cutting-edge technology.

Regarding to SI measure, please forgive us for not explaining clearly. We added some details in the Method section in the latest manuscript as below.

Signal intensity represent the magnitude of each pixel in a grayscale, and we follow this statement from previous studies on histogram features of intervertebral discs¹³. The histogram feature is used to quantify different signal intensity distribution in different areas from MRI, in which X-axis represents different signal intensities, and Y-axis represents the corresponding number of pixels. A two-peak distribution has been analyzed in healthy IVD from MRI, because the sharpness of the boundary between the NP and the AF can be well characterized with large amounts pixel with two major signal intensities (Fig. 5d)¹³. With IVD degeneration, water content loss in NP can be measured in histogram feature distribution changes, which presents that previous higher signal intensity (light) in the IVD area gradually becomes lower (dark) (Fig. 5g). The difference in pixel numbers corresponding to different signal intensities can well describe the degeneration state.

Reference

13. Waldenberg, C., Hebelka, H., Brisby, H. & Lagerstrand, K. M. MRI histogram analysis enables objective and continuous classification of intervertebral disc degeneration. *Eur. Spine J.* 27, 1042–1048 (2018).